# Role of syn-eruptive plagioclase disequilibrium crystallization in basaltic magma ascent dynamics

G. La Spina[1], M. Burton[1], M. de' Michieli Vitturi[2] & F. Arzilli[1]

Timescales of magma ascent in conduit models are typically assumed to be much longer than crystallization and gas exsolution for basaltic eruptions. However, it is now recognized that basaltic magmas may rise fast enough for disequilibrium processes to play a key role on the ascent dynamics. The quantification of the characteristic times for crystallization and exsolution processes are fundamental to our understanding of such disequilibria and ascent dynamics. Here we use observations from Mount Etna's 2001 eruption and a magma ascent model to constrain timescales for crystallization and exsolution processes. Our results show that plagioclase reaches equilibrium in 1–2 h, whereas ascent times were <1 h. Using these new constraints on disequilibrium plagioclase crystallization we also reproduce observed crystal abundances for different basaltic eruptions. The strong relation between magma ascent rate and disequilibrium crystallization and exsolution plays a key role in controlling eruption dynamics in basaltic volcanism.

[1] School of Earth and Environmental Sciences, The University of Manchester, Manchester M13 9PL, UK. [2] Istituto Nazionale di Geofisica e Vulcanologia, Sezione di Pisa, Pisa 56126, Italy. Correspondence and requests for materials should be addressed to G.L.S. (email: giuseppe.laspina@manchester.ac.uk).

Magma ascent dynamics in volcanic conduits play a key role in determining the eruptive style of a volcano[1-3]. The lack of direct observations inside the conduit means that numerical conduit models, constrained with observational data, provide invaluable tools for quantitative insights into complex magma ascent dynamics[4]. The highly nonlinear, interdependent processes involved in magma ascent dynamics require several simplifications when modelling their ascent. Indeed, initial conduit models assumed a single gas phase, isothermal conditions, strict gas–magma coupling and crystal-free magma[5,6]. Later models reduced these simplifications, introducing gas–magma separation[7-9], different volatile species[10,11] and a total crystal phase (not distinguishing the mineral species)[12-15]. Further developments modelled single crystal phases separately as microlites and phenocrysts[16-18]. Maintaining chemical equilibrium between melt, crystal and fluid phases, regardless of magma ascent rate, is commonly assumed[12,15,19]. Indeed, timescales of magma ascent in conduit models[4,19] are typically assumed to be much longer than crystallization[20,21] and gas exsolution[22,23]. However, it is now recognized that basaltic magmas rise fast enough[1,23] that the time available for crystal growth and volatile exsolution may not allow equilibrium to be achieved, producing significantly different magma rheology and eruptive behaviour compared with an equilibrium ascent. Thus, the quantification of timescales for crystallization and exsolution processes are fundamental to our understanding of such disequilibria and ascent dynamics.

In basaltic eruptions, plagioclase typically crystallizes at relatively shallow depths ($< 10\,km$)[20,24-26], recording information of the magma ascent in the last kilometres below the vent. Considering a basaltic system characterized by water-saturated conditions and temperature of $\sim 1,100\,°C$, plagioclase crystallizes from 50/75 MPa (2/3 km) to the surface[20,25-27], recording textural and chemical information of syn-eruptive conditions. Therefore, plagioclase can act as a sensitive indicator of relatively shallow disequilibrium processes. Plagioclase crystallization is dependent on dissolved water contents[1,28] and is therefore sensitive to disequilibrium degassing of volatiles.

Using observations from the Mount Etna's 2001 flank eruption (see Supplementary Fig. 1, courtesy of Dr Boris Behncke, INGV Osservatorio Etneo) and a one-dimensional (1D) multiphase multi-component steady-state model for magma ascent[4], we constrain timescales for crystallization and exsolution processes, showing that equilibrium crystal content is reached in about 2 h, whilst equilibrium gas exsolution is achieved in $< 5\,s$. Furthermore, we relate the amount of plagioclase in erupted products with magma ascent dynamics, finding that a high plagioclase contents implies residence time in a shallow reservoir, whilst a low plagioclase content indicates a fast vertical ascent from depth to the surface. Using these new constraints on timescales for crystallization and exsolution, we reproduce observations of crystal abundances for basaltic eruptions from Stromboli and Kilauea. Therefore, our results show that disequilibrium crystallization and exsolution play a key role on the ascent dynamics of basaltic magmas and, thus, they cannot be ignored when modelling and interpreting basaltic eruptions.

## Results

**Equilibrium versus disequilibrium processes.** To explore the role of plagioclase equilibrium/disequilibrium crystallization in basaltic eruptions, we considered, as a test case, the 2001 flank eruption at Mount Etna (eastern Sicily, Italy). A description of this eruption is reported in Supplementary Note 1. During this eruption, two different kinds of products were observed[29,30]:

a plagioclase-rich magma (16–23 vol.%), with a high crystal content (30–39 vol.%), erupted from the vent located above 2,600 m above sea level (hereafter upper vents, UV); and a plagioclase-poor magma (4–7 vol.%) with a low crystal content (15–23 vol.%) from the vent located below 2,600 m above sea level (hereafter lower vents, LV). Furthermore, in ref. 29 it is shown that the crystallinity of tephra and lava flow erupted from UV is similar, meaning that the lava flow products are not affected by post-eruption crystallization. Thus, it is reasonable to assume that also in the LV post-eruption crystallization did not occur (or it was negligible), implying that the crystals observed in the erupted products grew during ascent. For this reason, the presence of two different kinds of products with different plagioclase contents makes this eruption the perfect test case to investigate the non-equilibrium crystallization and exsolution processes and the relationship between plagioclase content and ascent dynamics of basaltic magmas.

To study in detail the non-equilibrium crystallization and exsolution, we use a 1D multiphase multi-component steady-state model for magma ascent[4], in which the main physical and chemical processes (such as crystallization, exsolution, rheological variations, outgassing, non-ideal gas behaviour and temperature changes) are calculated (see Methods section). In this model, three different crystal components (plagioclase, clinopyroxene and olivine) and two different volatile species (water and carbon dioxide) are taken into account. This model is the first non-equilibrium conduit model that account for the crystallization of different minerals. The equilibrium crystal contents for any pressure, temperature and dissolved water content are determined using alphaMELTS[31], and, as magma ascends, the equilibrium crystal assemblage evolves. By controlling the rate (through a characteristic time) at which the model responds to changes in equilibrium crystal content we can determine whether equilibrium crystal volume fractions are achieved throughout the ascent, or not. Similarly, equilibrium $CO_2$ and $H_2O$ contents are calculated using VolatileCalc[32] for the P, T profile of magma ascent, and the exsolution characteristic time determines how swiftly equilibrium gas exsolution is achieved. The finite-rate crystallization and volatile exsolution are taken into account through the equations (5)–(8) illustrated in the Methods section. The crystallization and exsolution rate are controlled, respectively, by the characteristic times $\tau^{(c)}$ (s) and $\tau^{(e)}$ (s). These parameters reflect the time required to reduce the difference between the actual and the equilibrium value to $e^{-1}$ ($\sim 37\%$) of the initial difference. This means that, if $\phi_0$ is the initial value of a physical parameter $\phi$, and $\phi^{eq}$ is the equilibrium value in response to a perturbation of the system, at the characteristic time $\tau$, we have $\phi(\tau) = \phi^{eq} + e^{-1}(\phi_0 - \phi^{eq})$. If these characteristic times are of the same order of magnitude or larger than the magma ascent time (which controls the decompression rate), disequilibrium processes will affect the ascent dynamics.

The characteristic times for crystallization and exsolution are functions of pressure, temperature, dissolved water content and bulk composition of the melt. Experimental constraints on these characteristic time functions are lacking. In ref. 33, de' Michieli Vitturi et al. presented numerical results from the investigation of several disequilibrium processes using a numerical 1D steady-state model applied to a rhyolitic eruption, validating the use of a constant finite-rate exsolution against the data from ref. 34. For this reason, in this work, as a first-order approximation of the non-equilibrium crystallization and exsolution in basaltic magmas, we assume that the characteristic times $\tau^{(c)}$ and $\tau^{(e)}$ are constant during the ascent and are the same, respectively, for each crystal phase and each volatile component.

We investigated equilibrium magma ascent behaviour (obtained by assuming instantaneous crystallization and

exsolution, that is, defining $\tau^{(c)}$ and $\tau^{(e)}$ to be $10^{-5}$ s), for several initial conditions (see Methods section). While the numerical results are in good agreement with observed UV products, the LV products cannot be reproduced (Table 1). We then examined finite-rate crystallization and exsolution processes to evaluate if the observed LV crystal assemblage could be reproduced. In Fig. 1 we compare the numerical results obtained assuming both instantaneous and finite-rate crystallization and exsolution. From the numerical simulations at equilibrium conditions we have derived an ascent time of $\sim 1$ h from a depth of 9 km, therefore by defining $\tau^{(c)}$ and $\tau^{(e)}$ to be $10^2$ s, disequilibrium crystallization and exsolution is produced. We observe a lower total crystal content than that achieved assuming equilibrium (23 versus 29 vol.% at the vent). Plagioclase is the most sensitive mineral to finite-rate processes, producing, at the vent, 17 and 5 vol.%, respectively, when instantaneous and finite-rate crystallization and exsolution are taken into account (Fig. 1e). The final pyroxene content shows an increase for disequilibrium ascent with respect to the equilibrium case (14 versus 9 vol.%), reflecting dissolution processes that take place during equilibrium ascent (Fig. 1e). Olivine content, instead, is not particularly affected by the finite-rate processes, resulting in 3 vol.% when instantaneous processes are considered, and in 4 vol.% when finite rates are taken into account (Fig. 1e). The crystal content computed from the numerical simulation at disequilibrium is in agreement with that observed during the LV activity (4–7 vol.% of plagioclase, 8–14 vol.% of pyroxene and 1–3 vol.% of olivine)[29], whilst numerical results at equilibrium are consistent with the UV products (16–23 vol.% of plagioclase, 10–19 vol.% of pyroxene and 1–3 vol.% of olivine)[29].

Disequilibrium processes have an important effect on the viscosity within the conduit (Fig. 1c). Lower plagioclase content and volatile supersaturation due to disequilibrium processes cause a decrease of the viscosity, resulting in a difference of almost an order of magnitude at the vent of the conduit (3,600 Pa s versus 500 Pa s). The eruption rates derived from the numerical simulations are $\sim 8\,m^3\,s^{-1}$ for the solution at instantaneous crystallization and exsolution, whilst $\sim 11\,m^3\,s^{-1}$ for that obtained assuming a finite rate, both in agreement with volcanological observations[35]. Furthermore, the gas volume fraction obtained assuming finite-rate crystallization and exsolution is larger (in the last 1.5 km below the vent) than that derived from the simulation at instantaneous condition. In particular, at the conduit mouth, we obtained 81 and 76 vol.% of exsolved gas at disequilibrium and equilibrium crystallization and exsolution, respectively. Thus, even though the model does not take into account fragmentation, if we

had considered a volume fraction criterion for the fragmentation of magma with a threshold at 80 vol.%, the solution at disequilibrium crystallization and exsolution would have produced an explosive eruption. Therefore, a variation in the magma ascent rate (for example, due to perturbations of the initial condition) could change the degree of disequilibrium crystallization and exsolution, producing, eventually, a change in the eruptive style. For this reason, feedback between disequilibrium processes and magma ascent rate could be a key factor controlling transitions in eruption style.

From the numerical results presented here, we have shown that while the characteristic times for crystallization and exsolution may vary during ascent, we are able to reproduce the observed characteristics of LV activity products, suggesting that characteristic times can be modelled accurately using 'effective', constant, characteristic times. Moreover, it is reasonable to assume that they do not depend on the plumbing system, and therefore the characteristic times used to describe LV should also hold for the UV. For this reason, since we are not able to reproduce the observations from the UV with a finite-rate crystallization and exsolution, a possible explanation is that the assumption of a vertical straight conduit is not valid for the UV. The equilibrium crystal content observed in UV can be explained with a longer ascent time than the LV, potentially due to a shallow residence in the upper plumbing system. Therefore, our results show that LV activity can be modelled with a non-equilibrium fast ascent from 9 km to the surface, whilst UV magmas suggest the presence of a shallow chamber in which magma can reside for sufficient time to reach equilibrium before being erupted. The high plagioclase content of UV products requires a low water content, and implies that this shallow reservoir should be located a few hundred metres below the vent. This is consistent with the prevailing literature, in which it is suggested that LV were fed by a straight vertical conduit ('eccentric' activity) whilst UV were fed a central conduit with a shallow horizontal dyke ('central-lateral' activity)[29,30,36–46]. Furthermore, from petrological evidence, the shallow horizontal dyke should be located between 75 and 5 MPa (refs 29,46), that is, between $\sim 2.5$ km and the vent of the conduit. Constraints from crystal assemblages therefore demonstrate that disequilibrium processes are key to explain the multivent behaviour of the 2001 flank eruption at Mount Etna.

**Sensitivity analysis.** Using Etna as a natural laboratory, we performed a sensitivity study to provide better constraints on

**Table 1 | Numerical results and observations from Etna 2001 flank eruption.**

| | | | | | | | | | | |
|---|---|---|---|---|---|---|---|---|---|---|
| | **Numerical results** | | | | | | | | | |
| | **Input** | | | | | **Output** | | | | |
| Crystallization model | $x_{d_{H_2O}}$ (wt.%) | $x_{d_{CO_2}}$ (wt.%) | $T_{chamb}$ (K) | $\tau^{(c)}$ (s) | $\tau^{(e)}$ (s) | $\beta_{plg}$ (vol.%) | $\beta_{cpx}$ (vol.%) | $\beta_{ol}$ (vol.%) | $\beta_{tot}$ (vol.%) | VFR ($m^3\,s^{-1}$) |
| LV | 3.4 | 0.4 | 1363 | $10^{-5}$ | $10^{-5}$ | 17 | 9 | 3 | 29 | 9 |
| UV | 3.4 | 0.4 | 1363 | $10^{-5}$ | $10^{-5}$ | 16 | 9 | 2 | 27 | 11 |
| LV | 6.4 | 0.4 | 1363 | $10^{-5}$ | $10^{-5}$ | 14 | 9 | 2 | 25 | 30 |
| LV | 3.4 | 0.4 | 1383 | $10^{-5}$ | $10^{-5}$ | 12 | 7 | 2 | 21 | 36 |
| LV | 3.4 | 0.4 | 1363 | $10^{2}$ | $10^{2}$ | 5 | 14 | 4 | 23 | 11 |
| | **Observations** | | | | | | | | | |
| | | | | | | **Field data** | | | | |
| Etna 2001—upper vents | | | | | | 16–23 | 10–19 | 1–3 | 30–39 | 1–30 |
| Etna 2001—lower vents | | | | | | 4–7 | 8–14 | 1–3 | 15–23 | 1–30 |

Comparison of the numerical output (respectively, plagioclase, clinopyroxene, olivine, total crystal content and volume flow rate, VFR) at the vent of the conduit for the Etna 2001 flank eruption obtained assuming different initial conditions and the observations from upper vents and lower vents. The calculations in equilibrium conditions are in agreement with observed UV products, while the LV products cannot be reproduced unless disequilibrium crystallization and exsolution are considered.

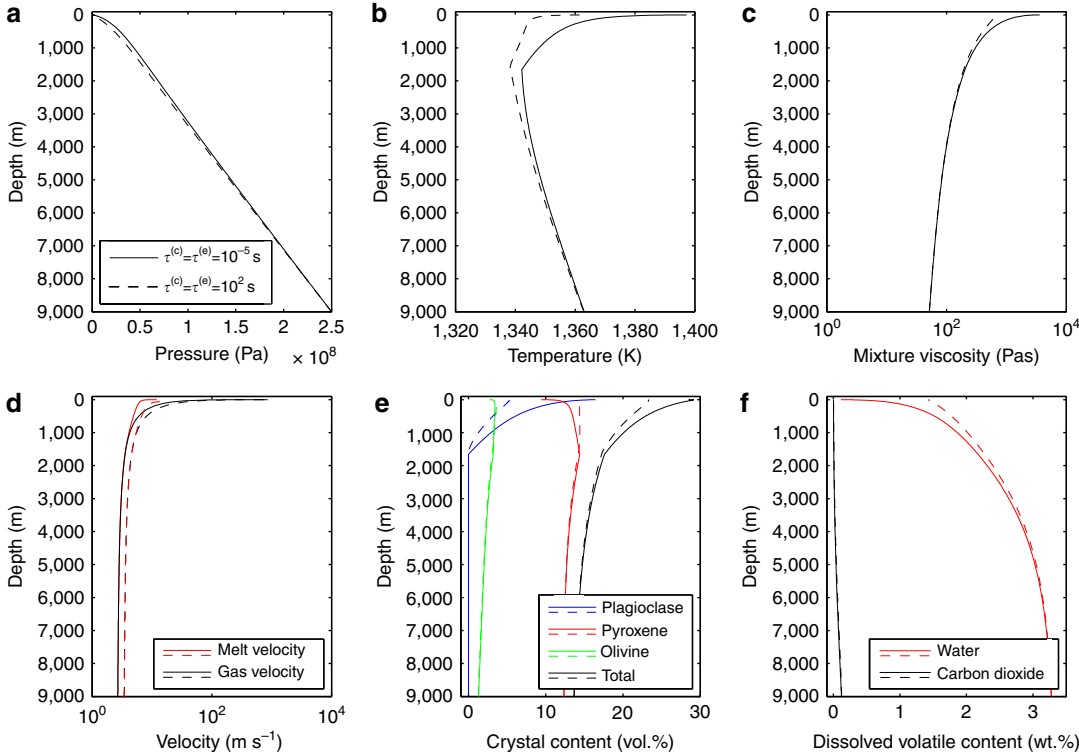

**Figure 1 | Numerical results at equilibrium and disequilibrium conditions.** Pressure (**a**), temperature (**b**), mixture viscosity (**c**), velocities (**d**), crystal contents (**e**) and dissolved volatile contents (**f**) as function of depth, computed using LV crystallization model and assuming, respectively, equilibrium (solid lines) and disequilibrium (dashed lines) crystallization and exsolution. We see that temperature decreases in the lower part of the conduit, while around 1.5 km of depth it starts to increase. The increase in temperature is related to the plagioclase crystallization that starts to nucleate at ~1.5 km of depth. The continuous crystallization of plagioclase, in fact, releases heat to the system increasing the temperature of the mixture. These variations in temperature have a great impact on the crystal content and on the viscosity during the ascent, since these are strictly controlled by the temperature. Finally, both disequilibrium processes produce a decrease in the temperature with respect to that obtained assuming equilibrium conditions, obtaining a difference in temperature at the vent of the conduit of ~35 K (1,363 versus 1,397 K).

characteristic times for crystallization and exsolution, values that are challenging to deduce from laboratory experiments. Assuming that $\tau^{(c)}$ and $\tau^{(e)}$ do not depend on the plumbing system, that is, they are the same for both LV and UV activities, we estimate the time required to obtain the crystal content observed from UV. In Fig. 2 we report the volume fractions of each crystal component and the total crystal volume fraction at the vent as functions of $\tau^{(c)}$ and $\tau^{(e)}$. Compared with ref. 29, we have considered a greater range of plagioclase content (3–8 vol.%) to take into account uncertainties in the calculated crystal contents derived from the parameterization of MELTS computations[4]. Furthermore, we have limited our region of admissibility assuming that exsolution is a process occurring faster than crystallization, that is, $\tau^{(e)} \leq \tau^{(c)}$. This is a reasonable assumption since results from ref. 47 show that vesiculation of basaltic magmas happens over timescales of tens of seconds, whilst data from refs 26,27 indicate that equilibrium crystallization is reached in about 2 h. Furthermore, the growth rates of vesicles ($10^{-4}$–$10^{-2}$ cm s$^{-1}$)[47,48] are several order of magnitude higher than those of plagioclase ($10^{-7}$–$10^{-8}$ cm s$^{-1}$)[26,27]. These data suggest that the timescale of exsolution is shorter than crystallization. With these assumptions (Fig. 2a), we deduce for $\tau^{(c)}$ an upper limit of about 20 min (with $\tau^{(e)} \leq 1$ s) and a lower limit of about 30 s (with $\tau^{(e)} = 30$ s). Moreover, from this range of $\tau^{(c)}$, it is possible to derive estimates of the crystallization time, that is, the time needed for plagioclase to reach equilibrium (see Methods section), obtaining a range from 2 (with a long

characteristic time for exsolution) to 100 min. Similarly, from the range of $\tau^{(e)}$, we calculate that the gas exsolution timescale is <3 min.

Laboratory experiments of plagioclase crystallization show a very wide range of timescales for reaching equilibrium volume fraction, going from 2 to 20 h (refs 20,21,27). Comparison with our numerical results suggests that a longer timescale for crystallization and a shorter timescale for gas exsolution are most realistic, and that ~100 min is required for plagioclase to reach equilibrium after a pressure change, with characteristic time of ~20 min. This timescale is faster than laboratory results, perhaps because the natural system includes the combination of several processes, which are not simulated in the experiments. Our results, instead, include all the main processes, which can control crystallization rate, such as variations in undercooling, viscosity, strain rate and degassing, making the quantification of the characteristic time for plagioclase crystallization more realistic. With a crystallization time of ~100 min, the gas exsolution timescale is limited to <5 s. A crystallization time of about 100 min is consistent with a shallow magma residence hypothesis for UV activity, in which fresh magma arrives in disequilibrium in a shallow reservoir, circulates within it for at least 1–2 h reaching the equilibrium crystal content and then rises through the upper part of the conduit again in disequilibrium conditions.

As stated above, characteristic times for crystallization and exsolution are likely functions of the pressure, temperature, dissolved water content and of the bulk composition of the melt,

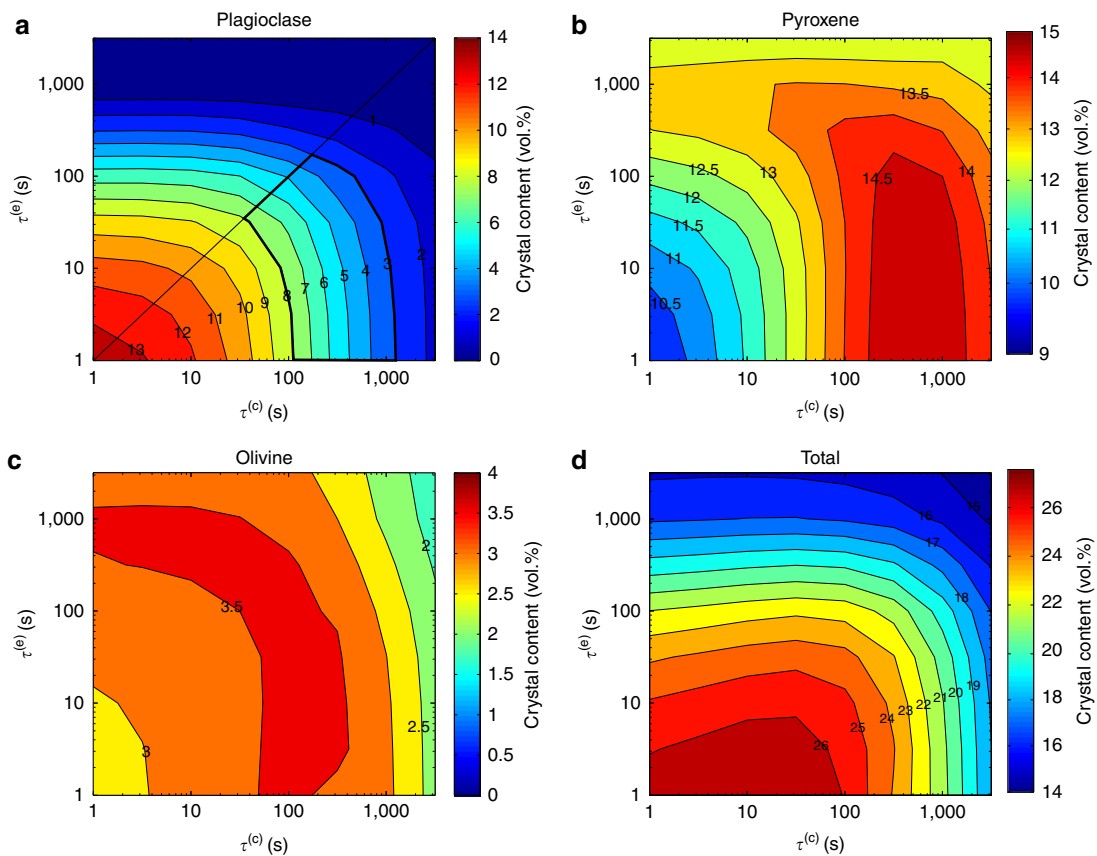

**Figure 2 | Sensitivity analysis on the characteristic times.** Contents of plagioclase (**a**), pyroxene (**b**), olivine (**c**) and total crystallinity (**d**) at the vent of the conduit as function of characteristic times for crystallization and volatile exsolution. Sensitivity analysis shows that plagioclase is the crystal phase affected most by disequilibrium processes, ranging from 0 vol.% (in the case of strong disequilibrium) to 14 vol.%. Pyroxene shows a smaller variability compared with plagioclase, ranging from a minimum 10 vol.% to a maximum of 15 vol.%. On the contrary, olivine is almost unaffected by disequilibrium processes, showing a content of 2–4 vol.%. The admissibility region for plagioclase has been obtained considering the range 3–8 vol.%, to take into account uncertainties in the calculated crystal contents derived from the parameterization of MELTS computations[4], and assuming that exsolution is a faster process than crystallization (that is, $\tau^{(e)} \leq \tau^{(c)}$).

and, therefore, they could vary with the different volcanic systems. However, for volcanic systems with a similar bulk composition it is reasonable to assume that the corresponding characteristic times are quite similar to each other, at least at the first order. Therefore, we have used the values for $\tau^{(c)}$ and $\tau^{(e)}$ deduced from Etna for other similar basaltic systems by performing numerical tests using magmatic compositions of Stromboli and Kilauea, whose products contain, respectively, high and low plagioclase contents. In Fig. 3 we report plagioclase content as function of the volume flow rate, obtained from numerical simulations of Etna, Stromboli and Kilauea. The initial conditions used are reported in the Methods section. The grey regions in Fig. 3, delimited by the two curves obtained by defining $\tau^{(c)}$ and $\tau^{(e)}$, respectively, as 1,200 and 1 s and 30 and 30 s, represent the areas in which the solutions are compatible with a fast, straight ascent in a vertical conduit. Above this region, a simple vertical magma ascent would be too fast to allow the crystallization of that plagioclase content, and, thus, we need a shallow reservoir in which plagioclase can crystallize for enough time and reach the proper content. In Fig. 3 we have also highlighted several areas representing observations from different eruptions that occurred at the considered volcanoes: Etna 2001 (refs 29,35) and 2002/2003 (ref. 38); Stromboli 2007 (effusive eruption)[4,49,50] and 1930 (paroxysm)[51]; and different Pu'u' O'o eruptions at Kilauea (episodes 49–53)[52]. Similarly to the Etna 2001 flank eruption, observations from

South fissure of Etna 2002/2003 lay just below the grey region (Fig. 3a) indicating a fast vertical ascent, whilst data from North fissure are above this region, indicating the presence of a shallow reservoir. In Fig. 3b, the high plagioclase content of samples collected during the 2007 effusive eruption at Stromboli[49] are in agreement with the equilibrium assumption, suggesting that magma crystallizes in a shallow reservoir before eruption. In contrast, the low plagioclase content of the 'golden' pumice PST-9 collected during the 1930 paroxysm at Stromboli[51] is consistent with a simple, fast, disequilibrium ascent in a vertical conduit. Finally, for Pu'u' O'o eruptions at Kilauea, the absence of plagioclase indicates a fast vertical ascent (Fig. 3c). The agreement of our results with eruption products and previous interpretations of the plumbing systems of different basaltic volcanoes allows us consider, at least at the first order, a constant characteristic time of ∼20 min for crystallization and <1 s for exsolution, as a general result for basaltic volcanism.

## Discussion
In conclusion, we have used a combination of basaltic eruption observations and numerical modelling to constrain the plagioclase crystallization characteristic time and the gas exsolution one, for several different basaltic compositions. Our results show that, after depressurization, equilibrium

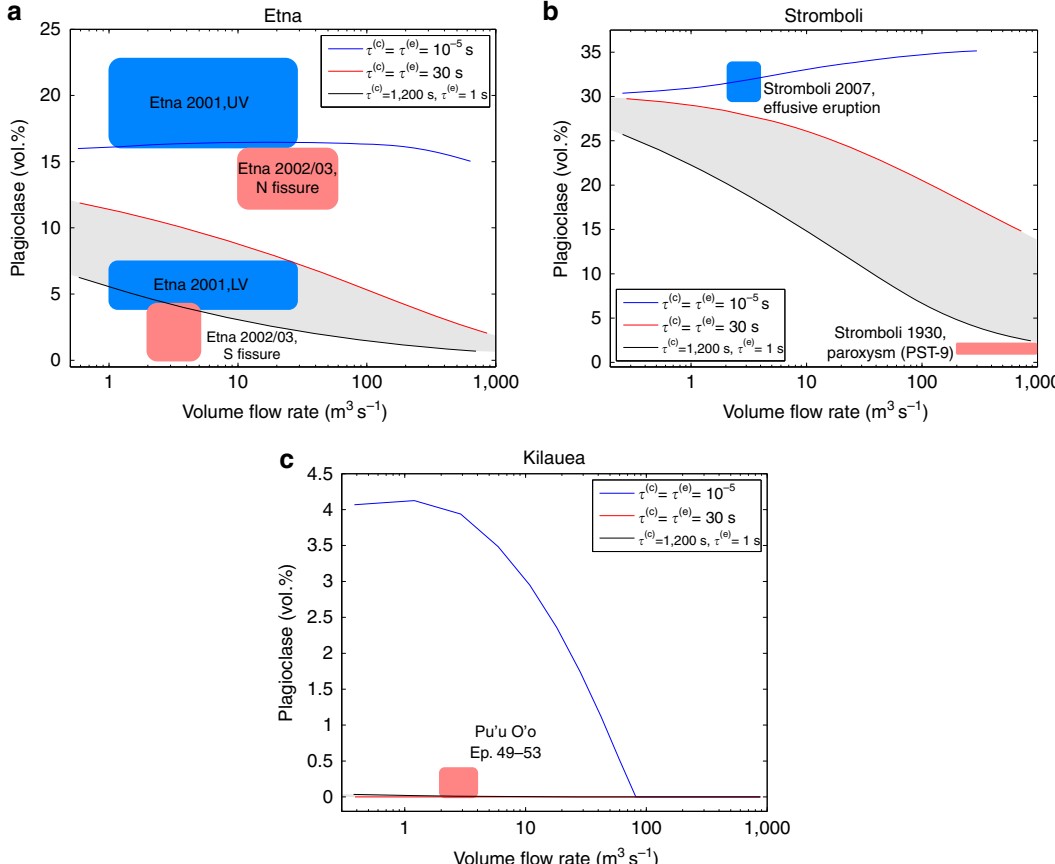

**Figure 3 | Numerical results for different basaltic volcanic systems.** Plagioclase content obtained from the numerical results at different volume flow rate, respectively, for Etna (**a**), Stromboli (**b**) and Kilauea (**c**). For each volcanic system, we have compared the numerical solutions computed assuming equilibrium crystallization and exsolution, that is, posing $\tau^{(c)}$ and $\tau^{(e)}$ to $10^{-5}$ s, with those calculated assuming the disequilibrium needed to obtain, respectively, the maximum and minimum plagioclase content showed in the previous sensitivity analysis on Etna, that is, defining $\tau^{(c)}$ and $\tau^{(e)}$, respectively, to 1,200 and 1 s and 30 and 30 s. The grey regions represent the areas in which we have a fast straight ascent from the bottom to the vent of the conduit. We have also reported several areas representing observations from different eruptions occurred at the considered volcanoes: Etna 2001 (refs 31,35) and 2002/2003 (ref. 38); Stromboli 2007 (effusive eruption)[4,49,50] and 1930 (paroxysm)[51]; and different Pu'u' O'o eruptions at Kilauea (episodes 49–53)[52]. With respect to the corresponding vents of the 2001 eruption of Etna, the plagioclase content observed in 2002/2003 is lower, suggesting also different initial conditions for this eruption (such as a higher temperature of the chamber or a different water content).

crystal content is achieved in ~100 min and equilibrium gas exsolution is achieved in <5 s. Furthermore, we have related the amount of plagioclase in erupted products with the ascent dynamics of basaltic eruptions. We find that relatively high plagioclase content requires crystallization in a shallow reservoir, whilst a low plagioclase content reflects a disequilibrium crystallization occurring during a fast ascent from depth to the surface.

The timescale of magma ascent and disequilibrium processes could control also the transition in eruptive styles, since an increase in magma ascent rate produces an increase in the gas volume fraction in the upper part of the conduit, resulting, eventually, in the fragmentation of magma. Furthermore, numerical results have shown that viscosity can vary by almost an order of magnitude between a full equilibrium eruption (3,600 Pa s) and a non-equilibrium eruption (500 Pa s) due to the impact of disequilibrium crystallization and exsolution. Thus, disequilibrium processes play a key role on the ascent dynamics of basaltic magmas and cannot be neglected when describing basaltic eruptions. Therefore, quantifying the characteristic times for crystallization and exsolution represents a major step towards a more complete, realistic and general model of basaltic volcanism.

## Methods

**Governing equations for the 1D steady-state conduit model.** The 1D steady-state model for magma ascent used in this work was presented in ref. 4, and it is designed to model multiphase flows with disequilibrium processes. The flow of the magmatic multiphase multi-component mixture along the $z$ axis is treated as a continuum and the state of the two phases, denoted by the index $k = l$, g, is characterized by its volume fraction ($\alpha_k$), mass density ($\rho_k$), velocity ($u_k$) and specific entropy ($s_k$). The first phase represents a 'liquid' phase, a mixture of melt, crystals and dissolved gas, while the second phase is a 'gas' phase, that is, the bubbles of exsolved gases. Thus, for the volume fractions, the saturation constraint $\alpha_l + \alpha_g = 1$ holds. Since the second phase is always referred to as the exsolved gas phase, we will use, without ambiguity, the subscripts $g_1$ and $g_2$ to refer to the two different gas species ($H_2O$ and $CO_2$). Furthermore, the subscripts $d_1$ and $d_2$ are used to refer to the dissolved gas species, the subscript m is used for the melt, while $c_1$, $c_2$ and $c_3$ are used to refer to the three different crystal phases (plagioclase, pyroxene and olivine).

Since the focus of our study is multiphase dynamics, separation of the gas phase from the liquid phase is permitted ($u_l \neq u_g$ where $u_l = u_m = u_{d_i} = u_{c_j}$ and $u_g = u_{g_i}$). Different pressures between the gas and the liquid phases can be also taken into account ($P_l \neq P_g$, where $P_l = P_m = P_{d_i} = P_{c_j}$ and $P_g = P_{g_i}$), while for the temperatures a condition of equilibrium between the phases is assumed ($T = T_m = T_{d_i} = T_{c_j} = T_{g_i}$). Although the model allows different pressures and velocities between gas and liquid phases, by tuning relaxation parameters it is also possible to impose an instantaneous pressure and velocity equilibrium between gas and liquid phases.

Following ref. 4, the steady-state 1D system of conservation equations is derived from the theory of thermodynamically compatible systems[53], where the conservation equations are expressed not as separate phases, but for the whole

mixture:

$$\frac{\partial \rho u}{\partial z} = 0, \tag{1}$$

$$\frac{\partial}{\partial z}\left[\sum_{k=\mathrm{l,g}} \alpha_k \rho_k u_k^2 + \alpha_k P_k\right] = -\rho g - \frac{8\mu_{\mathrm{l}} u_{\mathrm{l}}}{r^2}, \tag{2}$$

$$\frac{\partial}{\partial z}\left[\sum_{k=\mathrm{l,g}} \alpha_k \rho_k u_k\left(e_k + \frac{P_k}{\rho_k} + \frac{u_k^2}{2}\right) - \rho x_{\mathrm{l}} x_{\mathrm{g}}(u_{\mathrm{l}} - u_{\mathrm{g}})(s_{\mathrm{l}} - s_{\mathrm{g}})T\right] = -\rho g u - \frac{8\mu_{\mathrm{l}} u_{\mathrm{l}}^2}{r^2} \tag{3}$$

The above equations are, respectively, the conservation equation for the mixture density (equation (1)), the balance law for the mixture momentum (equation (2)) and for the mixture energy (equation (3)). Here we have defined the mixture density as $\rho = \alpha_{\mathrm{l}} \rho_{\mathrm{l}} + \alpha_{\mathrm{g}} \rho_{\mathrm{g}}$ and the mixture velocity as $u = x_{\mathrm{l}} u_{\mathrm{l}} + x_{\mathrm{g}} u_{\mathrm{g}}$, where $x_{\mathrm{l}}$ and $x_{\mathrm{g}}$ are, respectively, the mass fractions of the liquid and gas phases. Furthermore $g$ is the gravitational acceleration, $\mu_{\mathrm{l}}$ is the viscosity of the liquid phase, $r$ is the conduit radius, which is assumed to be constant, $e_{\mathrm{l}}$ and $e_{\mathrm{g}}$ are the specific internal energies, and $s_{\mathrm{l}}$ and $s_{\mathrm{g}}$ are the specific entropies. As in refs 15,19,54, the viscous terms included in the momentum and mixture energy equations are derived from the Poiseuille's approximation for 1D laminar flow.

The following balance equation, instead, rules the variations of the liquid volume fraction along the conduit:

$$\frac{\partial \rho u \alpha_{\mathrm{l}}}{\partial z} = -\frac{1}{\tau^{(\mathrm{p})}}(P_{\mathrm{g}} - P_{\mathrm{l}}). \tag{4}$$

The relaxation parameter $\tau^{(\mathrm{p})}$ ($\mathrm{m^2\,s^{-1}}$) controls the disequilibrium between the gas and liquid pressures. Here we assume equilibrium between the two pressures (that is, $\tau^{(\mathrm{p})} << 1$). In this way, as the fluid degases the pressure of the gas phase increases with respect to the liquid, and in response the liquid volume fraction is reduced to achieve liquid–gas pressure equilibrium.

The exsolution of each gas component and the corresponding dissolved contents are governed by the following balance equations

$$\frac{\partial \alpha_{\mathrm{g}_i} \rho_{\mathrm{g}_i} u_{\mathrm{g}}}{\partial z} = \frac{1}{\tau^{(\mathrm{e})}}\left(x_{\mathrm{d}_i}^{\mathrm{md}} - x_{\mathrm{d}_i}^{\mathrm{md,eq}}\right)\left(\alpha_{\mathrm{l}} \rho_{\mathrm{l}} - \sum_j \alpha_{\mathrm{l}} \rho_{\mathrm{c}_j} \beta_j\right), \tag{5}$$

$$\frac{\partial \alpha_{\mathrm{l}} \rho_{\mathrm{l}} x_{\mathrm{d}_i}^{\mathrm{md}} u_{\mathrm{l}}}{\partial z} = -\frac{1}{\tau^{(\mathrm{e})}}\left(x_{\mathrm{d}_i}^{\mathrm{md}} - x_{\mathrm{d}_i}^{\mathrm{md,eq}}\right)\left(\alpha_{\mathrm{l}} \rho_{\mathrm{l}} - \sum_j \alpha_{\mathrm{l}} \rho_{\mathrm{c}_j} \beta_j\right), \tag{6}$$

where $\beta_j$ is the volume fraction of the crystal component $j$, $x_{\mathrm{d}_i}^{\mathrm{md}}$ is the mass fraction of the dissolved gas phase $i$ with respect to the liquid crystal–free phase while $x_{\mathrm{d}_i}^{\mathrm{md,eq}}$ is the value at the equilibrium. The relaxation parameter $\tau^{(\mathrm{e})}$ (s) control the disequilibrium of the exsolution process.

The differential equation reported below governs the variations of the volume fractions for each crystal component:

$$\frac{\partial \alpha_{\mathrm{l}} \rho_{\mathrm{c}_j} \beta_j u_{\mathrm{l}}}{\partial z} = -\frac{1}{\tau^{(\mathrm{c})}} \alpha_{\mathrm{l}} \rho_{\mathrm{c}_j}\left(\beta_j - \beta_j^{\mathrm{eq}}\right). \tag{7}$$

Here $\beta_j^{\mathrm{eq}}$ is the volume fraction of the crystal component $j$ at the equilibrium. The relaxation parameter $\tau^{(\mathrm{c})}$ (s) in equation (7) controls the disequilibrium crystallization process.

Finally, the relative motion of the gas with respect the liquid phase is ruled by the following differential equation:

$$\frac{\partial}{\partial z}\left[\frac{u_{\mathrm{l}}^2}{2} - \frac{u_{\mathrm{g}}^2}{2} + e_{\mathrm{l}} + \frac{P_{\mathrm{l}}}{\rho_{\mathrm{l}}} - e_{\mathrm{g}} - \frac{P_{\mathrm{g}}}{\rho_{\mathrm{g}}} - (s_{\mathrm{l}} - s_{\mathrm{g}})T\right] = -\frac{1}{\tau^{(\mathrm{f})}}\frac{x_{\mathrm{l}} x_{\mathrm{g}}}{\rho}(u_{\mathrm{l}} - u_{\mathrm{g}}) - \frac{8\mu_{\mathrm{l}} u_{\mathrm{l}}}{\alpha_{\mathrm{l}} \rho_{\mathrm{l}} r^2} \tag{8}$$

The relaxation parameter $\tau^{(\mathrm{f})}$ ($\mathrm{kg^{-1}\,m^3\,s}$) controls the degree of decoupling between the gas and the liquid phase. Indeed, multiplying all terms of equation (8) by $\tau^{(\mathrm{f})}$, we can see that when this relaxation parameter is small enough we are forcing the relative velocity to be 0. On the contrary, the greater is $\tau^{(\mathrm{f})}$ the stronger will be the decoupling.

As we stated before, the disequilibrium crystallization and exsolution are controlled by the relaxation parameters $\tau^{(\mathrm{c})}$ (s) and $\tau^{(\mathrm{e})}$ (s). These coefficients are the characteristic times for their corresponding process. These parameters reflect the time required to reduce the difference between the actual and the equilibrium value to $e^{-1}$ of the initial difference, in response to a perturbation of the system. The smaller the characteristic time the faster the corresponding process converges towards its equilibrium state. From the definition of characteristic time, we derived that the minimum time needed by the crystals to reach at least 99% of the equilibrium volume fraction is about 5 times $\tau^{(\mathrm{c})}$. More details on this calculation are reported afterwards.

**Constitutive equations for the 1D steady-state conduit model.** To reproduce an eruptive activity with our numerical model, we need to define proper constitutive equations. These equations are used to describe specific processes for the considered volcano, such as magma rheology, solubility, crystallization and the decoupling between liquid and gas phases.

Following ref. 55, the viscosity of the liquid phase is modelled as

$$\mu_{\mathrm{l}} = \mu_{\mathrm{melt}} \cdot \theta\left(x_{\mathrm{c}}^{\mathrm{l}}\right), \tag{9}$$

where $\mu_{\mathrm{melt}}$ is the viscosity of the bubble-free, crystal-free liquid phase and $\theta$ is a factor that increases viscosity due to the presence of crystals[56].

We use an empirical relationship to estimate $\mu_{\mathrm{melt}}$ as a function of water concentration and temperature, as in ref. 57 (based on the Vogel–Fulcher–Tammann equation):

$$\log(\mu_{\mathrm{melt}}) = A + \frac{B\left(y, x_{\mathrm{d_{H_2O}}}^{\mathrm{md}}\right)}{T - C\left(y, x_{\mathrm{d_{H_2O}}}^{\mathrm{md}}\right)}, \tag{10}$$

where the viscosity $\mu_{\mathrm{melt}}$ is in Pa s, $T$ is the temperature in Kelvin, and $A$, $B$ and $C$ are appropriate parameters. The parameter $A$ is the logarithmic value of the viscosity at infinite temperature and it is assumed to be constant for all melts[57]. The parameters $B$ and $C$, instead, are functions of the bulk composition $y$ and of the dissolved water content $x_{\mathrm{d_{H_2O}}}^{\mathrm{md}}$.

Furthermore, as crystallization proceeds, viscosity is increased according to the empirical model described in ref. 58:

$$\theta = \frac{1 + \varphi^\delta}{[1 - F(\varphi, \xi, \gamma)]^{B\phi^*}}, \tag{11}$$

where

$$F = (1 - \xi)\mathrm{erf}\left(\frac{\sqrt{\pi}}{2(1 - \xi)}\varphi(1 + \varphi^\gamma)\right), \varphi = \frac{\left(\sum_{j=1}^{n_c} x_{\mathrm{c}_j}^{\mathrm{l}}\right)}{\phi^*}. \tag{12}$$

The fitting parameters $B$, $\delta$, $\xi$, $\gamma$ and $\phi^*$ chosen for this work are the same used in ref. 4: $\phi^* = 0.39$, $\gamma = 0.84$, $\xi = 0.03$, $B = 2.8$ and $\delta = 2 - \gamma$.

In this model, we consider two volatile components, $H_2O$ and $CO_2$, and the equilibrium profile of the dissolved gas content $x_{\mathrm{d}_i}^{\mathrm{md,eq}}$ of component $i$ follows the Henry's Law, that is

$$x_{\mathrm{d}_i}^{\mathrm{md,eq}} = \sigma_i\left(\frac{P_{\mathrm{g},i}}{\bar{P}}\right)^{\varepsilon_i}, \tag{13}$$

where $P_{\mathrm{g},i} = \alpha_{\mathrm{g}_i} P_{\mathrm{g}} \alpha_{\mathrm{g}}^{-1}$ is the partial pressure of the $i$-th gas component expressed

**Table 2 | Parameters for LV and UV crystallization model.**

| | LV crystallization model | | | UV crystallization model | | |
|---|---|---|---|---|---|---|
| | **Plagioclase** | **Pyroxene** | **Olivine** | **Plagioclase** | **Pyroxene** | **Olivine** |
| $\zeta_{j,1}$ | $-2.68 \times 10^{-9}$ | $-4.48 \times 10^{-9}$ | $-7.49 \times 10^{-10}$ | $-2.88 \times 10^{-9}$ | $-3.89 \times 10^{-9}$ | $-9.16 \times 10^{-10}$ |
| $\zeta_{j,2}$ | $-9.12 \times 10^{-6}$ | $-2.55 \times 10^{-5}$ | $3.86 \times 10^{-6}$ | $-9.00 \times 10^{-6}$ | $-2.65 \times 10^{-5}$ | $4.02 \times 10^{-6}$ |
| $\zeta_{j,3}$ | $-8.02 \times 10^{-3}$ | $-7.94 \times 10^{-3}$ | $7.37 \times 10^{-3}$ | $-9.98 \times 10^{-3}$ | $-7.70 \times 10^{-3}$ | $7.23 \times 10^{-3}$ |
| $\zeta_{j,4}$ | $-7.87 \times 10^{-8}$ | $6.06 \times 10^{-7}$ | $-3.75 \times 10^{-8}$ | $-4.62 \times 10^{-8}$ | $5.78 \times 10^{-7}$ | $-3.55 \times 10^{-8}$ |
| $\zeta_{j,5}$ | $-1.01 \times 10^{-3}$ | $-9.88 \times 10^{-4}$ | $3.40 \times 10^{-4}$ | $-1.05 \times 10^{-3}$ | $-9.78 \times 10^{-4}$ | $3.39 \times 10^{-4}$ |
| $\zeta_{j,6}$ | $-7.10 \times 10^{-6}$ | $1.14 \times 10^{-5}$ | $-2.31 \times 10^{-6}$ | $-5.70 \times 10^{-6}$ | $1.12 \times 10^{-5}$ | $-2.23 \times 10^{-6}$ |
| $\zeta_{j,7}$ | $1.15 \times 10^{-4}$ | $-6.13 \times 10^{-4}$ | $4.85 \times 10^{-5}$ | $7.83 \times 10^{-5}$ | $-5.85 \times 10^{-4}$ | $4.68 \times 10^{-5}$ |
| $\zeta_{j,8}$ | $1.76 \times 10^{-2}$ | $5.39 \times 10^{-2}$ | $-9.81 \times 10^{-3}$ | $1.74 \times 10^{-2}$ | $5.61 \times 10^{-2}$ | $-1.01 \times 10^{-2}$ |
| $\zeta_{j,9}$ | $9.43 \times 10^{-1}$ | $1.03 \times 10^{0}$ | $-4.14 \times 10^{-1}$ | $1.00 \times 10^{0}$ | $1.02 \times 10^{0}$ | $-4.12 \times 10^{-1}$ |
| $\zeta_{j,10}$ | $-8.03 \times 10^{0}$ | $-2.83 \times 10^{1}$ | $6.19 \times 10^{0}$ | $-7.99 \times 10^{0}$ | $-2.95 \times 10^{1}$ | $6.35 \times 10^{0}$ |

Fitting coefficients calculated over a large range of data obtained at different pressures, temperatures and water contents with alphaMELTS[31] using the whole-rock compositions, respectively, of sample 260701C (representative of the LV activity) and 240701D (representative of the UV activity)[29].

in Pa, $\bar{P} = 1$ Pa is used to make the expression in the brackets adimensional, $\sigma_i$ is the solubility coefficient and $\varepsilon_i$ is the solubility exponent. We assume that the solubility parameters $\sigma_i$ and $\varepsilon_i$ are constant during the ascent.

To obtain realistic values of the equilibrium profiles for basaltic magmas, we use the software VolatileCALC[32]. This code provide us with the individual saturation curves for water and carbon dioxide, to which we have applied a best fitting technique to obtain the parameters $\sigma_i$ and $\varepsilon_i$ of equation (13). The solubility curves for water and carbon dioxide are obtained assuming 49 wt.% of $SiO_2$ and a constant temperature of 1,100 °C, giving as results the following parameters[4]: $\sigma_{H_2O} = 5.0025 \times 10^{-7}$; $\varepsilon_{H_2O} = 0.6$; $\sigma_{CO_2} = 7.5563 \times 10^{-13}$; $\varepsilon_{CO_2} = 1.1$.

In this model, we consider three different crystal components: plagioclase; olivine; and pyroxene. We assume that crystals stay coupled with the melt (that is, no fractional crystallization). For a better modelling of crystal nucleation and growth, we also assume that the equilibrium crystal contents are functions of temperature, pressure and dissolved water content. With these assumptions, the equilibrium volume fraction $\beta_j^{eq}$ of crystal phase $j$ is computed using the polynomial function

$$\beta_j^{eq}(P^*, T^*, x_d^*) = \frac{\rho_1}{\rho_{c_j}} \left[ \zeta_{j,1}(P^*)^2 + \zeta_{j,2}(T^*)^2 + \zeta_{j,3}(x_d^*)^2 + \zeta_{j,4}(P^*)(T^*) + \right.$$
$$\left. + \zeta_{j,5}(T^*)(x_d^*) + \zeta_{j,6}(x_d^*)(P^*) + \zeta_{j,7}(P^*) + \zeta_{j,8}(T^*) + \zeta_{j,9}(x_d^*) + \zeta_{j,10} \right] \quad , \quad (14)$$

where $P^*$ is the liquid pressure expressed in bars, $T^*$ is the temperature expressed in °C and $x_d^*$ is the dissolved water concentration in wt.%. The parameters $\zeta_{j,i}$ are calculated fitting the polynomial function (equation (14)) over a large range of data obtained at different pressures, temperatures and water contents with alphaMELTS[31], a command line version of MELTS[59]. The two crystallization models adopted in the numerical simulations for the Etna's 2001 flank eruption are obtained from the different fitting coefficients calculated using the whole-rock compositions, respectively, of sample 260701C (representative of the LV activity) and 240701D (representative of the UV activity)[29]. The fitting coefficients for both crystallization models are reported in Table 2.

In the numerical model presented here, the outgassing is implemented following the Darcy's law through the relaxation parameter $\tau^{(f)}$ in the following form[4]:

$$\tau^{(f)} = \left( \frac{x_1^2 x_g^2}{\alpha_1 \alpha_g} \right) \left( \frac{k_{Darcian}}{\mu_g} \right). \quad (15)$$

In this expression $k_{Darcian}$ is the darcian permeability of the magmatic mixture, while $\mu_g$ is the viscosity of the gas phase. The darcian permeability can calculated as done in ref. 19:

$$k_{Darcian} = \frac{(f_{tb} r_b)^2}{8} \alpha_g^m. \quad (16)$$

In equation (16), $f_{tb}$ is the throat-bubble size ratio, $m$ is the tortuosity factor and $r_b$ is the average bubble size, which can be determined from the bubble number density and the gas volume fraction according to

$$r_b = \left( \frac{\alpha_g}{\frac{4\pi}{3} N_d \alpha_1} \right)^{\frac{1}{3}}, \quad (17)$$

where $N_d$ is the bubble number density. For this work we have posed $N_d = 10^9$ m$^{-3}$, $f_{tb} = 0.8$ and $m = 6$.

For melt, crystals and dissolved gas phases, a linearized version of the Mie–Grüneisen equations of state[4,60,61] is adopted:

$$e_k(\rho_k, T) = \bar{e}_k + c_{v,k} T + \frac{\rho_{0,k} C_{0,k}^2 - \gamma_k P_{0,k}}{\gamma_k \rho_k},$$
$$P_k(\rho_k, T) = c_{v,k}(\gamma_k - 1)\rho_k T - \frac{\rho_{0,k} C_{0,k}^2 - \gamma_k P_{0,k}}{\gamma_k},$$
$$s_k(\rho_k, T) = s_{0,k} + c_{v,k} \ln\left[ \frac{T}{T_{0,k}} \left( \frac{\rho_{0,k}}{\rho_k} \right)^{\gamma_k - 1} \right], \quad (18)$$

where $k = m$, $c_j$, $d_i$ for $j = 1, 2, 3$ (plagioclase, pyroxene and olivine) and $i = 1, 2$ (water and carbon dioxide). Here $\bar{e}_k$ is a constant parameter representing the formation energy of the fluid, $c_{v,k}$ is the specific heat capacity at constant volume, $\gamma_k$ is the adiabatic exponent, $\rho_{0,k}$, $P_{0,k}$, $T_{0,k}$, $s_{0,k}$ and $C_{0,k}$ are, respectively, the density, the pressure, the temperature, the specific entropy and the speed of sound at a reference state.

On the contrary, for the exsolved gas components, the van der Walls equations of state are adopted to take into account the non-ideality of gas phase[4]:

$$e_{g_i}(\rho_{g_i}, T) = c_{v,g_i} T - a_{g_i} \rho + \bar{e}_{g_i},$$
$$P_{g_i}(\rho_{g_i}, T) = c_{v,g_i}(\gamma_{g_i} - 1) T \frac{\rho_{g_i}}{1 - b_{g_i} \rho_{g_i}} - a_{g_i} \rho_{g_i}^2,$$
$$s_{g_i}(\rho_{g_i}, T) = c_{v,g_i} \log\left[ \frac{T}{T_{0,g_i}} \left( \frac{\rho_{0,g_i}}{\rho_{g_i}} \cdot (1 - b_{g_i} \rho_{g_i}) \right)^{(\gamma_{g_i} - 1)} \right], \quad (19)$$

for $i = 1, 2$. The coefficients $a_{gi}$ and $b_{gi}$ are defined as

$$a_{g_i} = \frac{27}{64} \frac{c_{v,g_i}^2 (\gamma_{g_i} - 1)^2 T_{c,g_i}^2}{P_{c,g_i}}, \quad b_{g_i} = \frac{1}{8} \frac{c_{v,g_i} (\gamma_{g_i} - 1) T_{c,g_i}}{P_{c,g_i}}, \quad (20)$$

where $P_{c,gi}$ and $T_{c,gi}$ are, respectively, the critical pressure and temperature of the gas component $g_i$ (ref. 62).

Owing to the stiffness of the terms included in the model, an implicit treatment and thus an accurate solution of the resulting nonlinear system is required. The solution of the model is computed adopting a shooting method to derive a magma input rate such that the pressure at the top of the conduit is equivalent to atmospheric pressure[63].

**Initial conditions for the numerical simulations.** For numerical eruption simulations of the 2001 flank eruption at Mount Etna, we have considered a cylindrical conduit of 9,000 m depth[29,30,64]. The magma chamber pressure and temperature are, respectively, 250 MPa and 1,363 K, while water and carbon dioxide contents are, respectively, 3.4 and 0.4 wt.% (ref. 30). For the equilibrium versus disequilibrium test case and for the sensitivity analysis on the characteristic times we have fixed the radius of the conduit to 1 m. This value has been obtained with the same procedure used in ref. 4, that is, performing several numerical simulations with different radius and comparing the resulting volume flow rate and the total crystal content with the real observations. In this way, we have found that, with a radius of 1 m and assuming instantaneous crystallization and exsolution, we obtain a solution in agreement with UV observations, whilst assuming finite-rate conditions, the corresponding solution is in agreement with the LV ones. In the last test, instead, where we have applied the characteristic times also to describe Stromboli and Kilauea eruptions, numerical results for Etna are obtained using different radii, from 0.5 to 3.0 m. Indeed, varying the radius of the conduit, the volume flow rate of the solution changes accordingly, allow us to relate the plagioclase content to the different volume flow rate.

For numerical simulations of Stromboli, instead, we have considered a cylindrical conduit of 10,000 m depth[4]. The magma chamber pressure and temperature are, respectively, 250 MPa and 1,343 K, while water and carbon dioxide contents are, respectively, 3.0 and 2.0 wt.% (refs 4,65). As we have done for the Etna in the last test presented in the manuscript, numerical results are obtained using different radii, from 0.5 to 3.0 m.

Finally, for numerical simulations of Kilauea, we have considered a cylindrical conduit of 3,000 m depth[52]. The magma chamber pressure and temperature are, respectively, 85 MPa and 1,433 K (ref. 52), while water and carbon dioxide contents are, respectively, 0.6 and 0.04 wt.% (ref. 66). Numerical results for Kilauea reported in the last test are obtained using different radii, from 0.3 to 2.5 m.

**Derivation of the crystallization time.** To estimate the time needed by the crystals to reach the equilibrium, we consider the time-dependent version of equation (7) that is

$$\frac{\partial}{\partial t}\left( \alpha_1 \rho_{c_j} \beta_j \right) + \frac{\partial}{\partial z}\left( \alpha_1 \rho_{c_j} \beta_j u_1 \right) = -\frac{1}{\tau^{(c)}} \alpha_1 \rho_{c_j} \left( \beta_j - \beta_j^{eq} \right). \quad (21)$$

We want to simulate the crystal growth (in terms of volume fraction) due to a sudden change in pressure and temperature. Therefore, assuming static condition and no changes in the crystal density, we can simplify the previous equation obtaining

$$\frac{\partial}{\partial t}\left( \alpha_1 \beta_j \right) = -\frac{1}{\tau^{(c)}} \alpha_1 \left( \beta_j - \beta_j^{eq} \right), \quad (22)$$

where $\alpha_1 \beta_j$ is crystal volume fraction with respect the whole mixture (liquid + exsolved gas phase). If we also assume that the volume fraction of the liquid is not changing during the experiments (that is, there is no exsolution occurring) we have

$$\frac{\partial}{\partial t}\left( \beta_j \right) = -\frac{1}{\tau^{(c)}} \left( \beta_j - \beta_j^{eq} \right). \quad (23)$$

This differential equation governs the variation in time of the crystal volume fraction and it can be used to provide constraint on the characteristic time $\tau^{(c)}$ through laboratory experiments. The solution of the previous equation, in fact, can be computed analytically and it is

$$\beta_j(t) = \beta_j^{eq} - \left( \beta_j^{eq} - \beta_j^0 \right) e^{-\frac{t}{\tau^{(c)}}}. \quad (24)$$

Therefore, using this formulation for the volume fraction of the crystals as function of time, it is possible to derive, given the characteristic time $\tau^{(c)}$, the time needed by the crystals to reach the equilibrium volume fraction $\beta_j^{eq}$. To have

$$\frac{\beta_j^{eq} - \beta_j(t)}{\beta_j^{eq} - \beta_j^0} < 0.01 \quad (25)$$

we need a time $t > 4.6\tau^{(c)}$.

**Computer code availability.** The computer code used within this study is available from the corresponding author on request

**Data availability.** The data that support the findings of this study are available from the corresponding author on request.

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

## Acknowledgements

We gratefully acknowledge funding support from RCUK NERC DisEqm project (NE/N018575/1). The research leading to these results has received funding from the European Research Council under the European Union's Seventh Framework Programme (FP/2007–2013)/ERC Grant Agreement no. 279802. This work was partly funded by the European Science Foundation (ESF), in the framework of the Research Networking Program MeMoVolc (M.d.M.V.). We are also grateful to Stephen J. Lane, Oleg Melnik and Chris Huber for their careful review and useful and constructive comments, which greatly improved the earlier version of this work.

## Author contributions

All authors contributed to production of the manuscript. G.L.S. performed the numerical simulations; F.A. contributed to the petrological elements of the paper.

## Additional information

**Competing financial interests:** The authors declare no competing financial interests.

