## [Peer Review File · Nature Communications]

Reviewer #1 (Remarks to the Author):

Summary of the key results.

The key result of identifying dis-equilibrium silicate melt with respect to volatile and crystal content driven by relatively rapid eruption rate is important. The results of the numerical simulations are tested against the properties of erupted materials and are consistent with existing hypothesised eruption scenarios and plumbing system structures.

Originality and interest.

This work is interesting and original, representing (to my knowledge) the first instance of coupling flow to phase (dis)equilibria: an important contribution.

Data & methodology: validity of approach, quality of data, quality of presentation.

The approach appears valid. It was not clear to me just how coupled the flow model was to the system disequilibrium and some clarification here would be useful. For instance, if the volatile phase was supersaturated was there less flow driving force?

The data are of high quality and the presentation good. I make comments on the attached annotated files.

Appropriate use of statistics and treatment of uncertainties.

A statistical approach is probably not appropriate here. Uncertainties are likely to be significant and are discussed. Nevertheless, the uncertainties appear small enough to allow discrimination between equilibrium systems and systems with significant dis-equilibrium.

Conclusions: robustness, validity, reliability.

The conclusions are robust in terms of demonstrating equilibrium from dis-equilibrium behaviour. However, I am concerned that the link to the 'effusive explosive transition' is stated but not discussed or demonstrated. This is important because it is the 'handle' on which the paper is sold, but the key findings remain important.

Suggested improvements: experiments, data for possible revision.

References: appropriate credit to previous work?

Clarity and context: lucidity of abstract/summary, appropriateness of abstract, introduction and conclusions.

I have annotated my detailed comments on to the attached pdf file.

SJL

Reviewer #2 (Remarks to the Author):

A. Summary of the key results

The paper under review considers disequilibrium crystallization of basaltic magma during ascent of the conduit. The main outcome of the modelling is that in order to explain Etna's product crystallinity a shallow reservoir located a few hundred meters below the summit should be considered. This statement is not really well supported by the conduit flow model (see below).

B. Originality and interest: if not novel, please give references

Results are original and interesting in a sense that multiple crystal species are involved in the simulation. There was no previous non-equilibrium conduit model that accounts for several minerals crystallization. Authors completely ignore papers by Melnik, Sparks and Costa that established the link between ascent rate and crystallinity during lava dome building eruption. The transition between explosive and effusive eruption is not discussed in the paper at all. Key references to Slezin (1983 and 2003) are missing. There is no link between proposed crystal growth kinetics and experimental data on crystal growth in basaltic magmas.

C. Data & methodology: validity of approach, quality of data, quality of presentation

The model assumes single characteristic disequilibrium timescale for all crystal types and for all P-T condition. This assumption cannot be justified both experimentally and theoretically. Magmatic undercooling that drives crystallization is a complicated function of magma ascent P-T-t path and residual melt composition. Assumption of the constant delay time oversimplifies the system significantly. Thus, as a theoretical approach the model is valid but it cannot lead to conclusive data interpretation.

D. Appropriate use of statistics and treatment of uncertainties

Some parametric study is performed in the paper but simple variation of τ_c and τ_d does not lead to proper kinetic description of crystallization.

E. Conclusions: robustness, validity, reliability

The title of the paper is misleading. The paper is not about the transition between different eruption styles but about syn-eruptive crystallization.

F. Suggested improvements: experiments, data for possible revision

Crystal growth model should be significantly reworked based on available non-equilibrium crystal growth experiments.

G. References: appropriate credit to previous work?

See above

H. Clarity and context: lucidity of abstract/summary, appropriateness of abstract, introduction and conclusions

The abstract is not strongly linked with the main body of the paper.

I suggest major revision of the manuscript.

Oleg Melnik

Reviewer #3 (Remarks to the Author):

Review of La Spina et al. : Crystallisation rate controls hazardous effusive-explosive transitions in basaltic volcanism

This manuscript presents a kinetic framework to study the influence of disequilibrium crystallization and exsolution on the product (crystal abundances) and eruption style at Etna and some comparisons with Stromboli and Kilauea. The paper is generally easy to read and follow and pleasant to read. There are some very good ideas behind this study that I would like to see discussed and published. On the other end, there are some, for the most part cosmetic, suggestions I have to improve the experience for the readers and also one or two scientific concerns that I think can all be reasonably addressed by the authors. In the next paragraphs, I start with the positive points and then discuss some current aspects I am not entirely convinced about and finally suggest some more detailed changes for the authors.

The most interesting and significant contribution of the authors to me is the use of two independent variables to constrain a simple steady-state multiphase (1-D) conduit flow model, more specifically to constrain the ascent time of the magma and its departure from equilibrium. The kinetics of volatile exsolution and crystallization is assumed here to operate over slightly different time scales, which provides two constraints on the conduit dynamics for the ascent of magmas where information about glass water content (or vesicularity) ADD CAVEAT THAT FOR BASALTIC FLOWS>>> VESICULARITY AND CRYSTAL CONTENT CAN BE AFFECTED BY POST ERUPTION !! and crystal content (and assemblage) can be used to "invert" or fit for the best ascent scenario. This is a clever and elegant idea, the model proposed is simple and promising, although several underlying assumptions and implication of these assumptions are either glossed over or not really discussed. The discussion of the effect of the degree of disequilibrium on the phase assemblage is interesting and although not a new concept, the model offers some nice ways to test some interesting ideas about these eruptions. The authors have been working on related issues for a while and have shown in many instances that they produce high quality work.

I have some reserves first of all about the validity of the first order kinetics assumed for the exsolution and crystallization in the mass balance. Is there a way to validate or justify this assumption outside of the fact that it simplifies the analysis? Moreover, I would have guessed that the "time constants" that the authors introduce in these first order reactions are probably not constants, but that the rate constants would depend on p, T and the magma composition, possibly even the flow conditions for exsolution (stretched bubbles have a greater Surface to Volume ratio so prone to grow faster). Because much of the paper revolves around the idea of these simple kinetic laws, it is worth spending more time discussing and justifying these choices.

The second point that needs to be discussed further is the constraint used to match or fit the most likely ascent scenario for each type of eruptions. Basaltic eruption products are known to be potentially affected by post-eruption processes, such as deformation upon landing and bubble collapse and partial volatile outgassing for explosive eruptions and crystallization and outgassing for

lava flows. Are the constraints used for the analysis corrected for these effects? If so how? The third point concerns the UV eruption products and the short storage required at shallow depth to maintain close to equilibrium conditions at low pressure before the last ascent. The physical model that the author use does not allow for a "reservoir" perched in the middle or higher level of the conduit. If I understand well the model can play with different (but constant over depth) time constant and ascent rates, but most likely not with a region where the magma stalls. It is therefore misleading to show that the choice of time constants allows the authors to discuss the cases where the crystal content is high and a shallow storage is believed to exist. The authors also added a constraint on the time constant by assuming that the time for degassing is shorter than the time for crystallization, a justification for what seems mostly plausible is required here.

Detailed comments:

- Title, although the effusive-explosive transition is part of the story, the title over-emphasizes the relevance of the present study to that question. The discussions with the viscosity feedbacks and other relevant aspect of the transition are relatively minor and not really the point of the paper. I would change the title.
- The reference to "time constants" knowing that the fact they are constants is merely an assumption is a bit weird. Could you refer to them as first order kinetic rates ($1/\tau$)? I think it is more clear and more precise.
- Be careful in the log-log figures, your arguments have units (seconds for the time constants), this is not really clean. Could you remove dimensions before or at least mention that the units on these graphs are log(sec) - which is a bit weird but at least it is a bit cleaner.
- The radius of the conduit at Etna was chosen as 1 meter, that is very narrow... any justification for that (all the way down the 9kms)? What radius did you pick for Stromboli and Kilauea?
- Line 124, you discuss how your model includes appropriate strain rate... well it is highly parameterized in a 1-D conduit model, you have some rough constraint on normal strain rates to some extent, but the shear rate not so much. I would be careful with that statement. Also the realism of your plag crystallization model relies on the realism of 1st order kinetics, which is not really discussed (see comment above).
- Lines 133-136: Any geophysical or other evidence for the stalling of magmas at shallow depth in its path to the surface?
- The model and how it is setup is discussed very briefly and lacks details. I found the supplements on that account to fall a bit short. The discussion of the e-folding time for τ 's is trivial but also requires on assumptions that are not relevant to the study (static magma...). That discussion to me should be replaced with a description of the model, initial and boundary conditions and sensitivity with initial conditions which are mostly missing. It would make the paper much more self-contained and pleasant to read.
- Why are the time constants fairly generic, i.e. applicable to other systems? The authors argue that it is controlled mostly by chemistry (thermodynamics), but the discussion there is a bit too succinct.
- Line 185-191: if you need some space, most of the sentences there could be removed or shortened. Hopefully, the readers should understand the difference between governing and constitutive equations.
- Line 195: how is the permeability parameterized and how does it affect your results? Much outgassing in your runs? There are no discussion on that anywhere else in the manuscript. I would write down the overall mass conservation for water (both dissolved and exsolved) or add the

contribution of the mass balance for the exsolved part to illustrate that.

- Line 218 and other places, please do not use "37%" in the text, which sounds arbitrary, rather used $1/e$.

Final remarks: I enjoyed this manuscript and I think that if the authors can make respond and add some justifications about their model and address the main comments (plus cleanup some of the details pointed just above), it will be worthy of publication in this journal. It was fun reading it, and reposed on several clever ideas.

Chris Huber

We thank SJL, Oleg Melnik and Chris Huber for all their comments. We have carefully revised the paper in light of their comments and we believe that following their suggestions we have significantly improved this manuscript.

One concern raised by the reviewers is the link made between magma ascent rate and the transition in eruptive styles. This was not one of the major results of the paper and for this reason, following the suggestion of all the reviewers, we have modified the title and the abstract of the manuscript, focussing more on the key findings of the paper. In any case, in the replies we have provided below, we show, comparing the results at equilibrium and disequilibrium crystallisation and exsolution, that we have an increase in the gas volume fraction in the solution at disequilibrium compared to that at equilibrium. Furthermore, a variation in the volume flow rate (for example due to a perturbation of the initial conditions) would produce a change in the degree of disequilibrium, resulting in a change of the gas volume fraction and, eventually, in the fragmentation of magma. Therefore, our results suggest that crystallisation plays a fundamental role in controlling transitions in eruptive style, but we understand that, at this time, we do not have enough evidence to strongly support this hypothesis. Thus, following the reviewers' suggestions, we have focussed the attention more on the role of plagioclase disequilibrium crystallisation in constraining the characteristic time for crystallisation and exsolution during ascent of basaltic magmas using the first non-equilibrium conduit model that account for the crystallisation of different minerals.

Several comments regard also the lack of details about the conduit model used to perform the numerical simulations. Following the suggestions of the reviewers we have reported, in the Methods Section, the complete description of the governing and constitutive equations of the model, in order to make it much more self-contained.

Other concerns raised by the reviewers regards the choice of constant characteristic times (or time constants). We agree with the reviewers that they are likely not to be constant, and a more accurate description should account for the effects of pressure, temperature, water content and bulk composition of the melt. However, de' Michieli Vitturi et al. (2013) presented numerical results from the investigation of several disequilibrium processes using a numerical 1D steady-state model applied to a rhyolitic eruption, validating the use of a constant finite-rate exsolution against the data from Mangan and Sisson (2000). Therefore, as a first order approximation to deal with disequilibrium crystallisation and exsolution, we have assumed that these characteristic times would be constant and the same respectively for each crystal phase and volatile component. The fact that we were able to reproduce the products observed from the LV suggests that our choice of characteristic times cannot be too far from the reality. Probably, during the ascent, the characteristic times vary all along the conduit going from a larger value at depth (where we have higher temperatures, small velocities, higher pressures), to a smaller value approaching to the vent (where we have lower temperatures, higher velocities and lower pressures). Our results suggest that the effects of the real non-constant characteristic times can be reproduced assuming sort of constant "effective" characteristic times, balancing their variation along the conduit.

Finally, another concern regards our interpretation on the presence of a shallow chamber below the UV. Using the 1D conduit flow model we have derived estimations for the characteristic times of crystallisation and exsolution, which give us solutions in agreement with the LV observations. Then, assuming that these characteristic times are a property of the magma composition and not of the plumbing system, we have concluded that these estimations should be valid also for the UV. However, the numerical results obtained assuming finite-rate crystallisation and exsolution are not in agreement with the UV observations (in particular due to the plagioclase content). Assuming that the model is describing properly the ascent dynamics of the eruptions

(since we were able to reproduce correctly the LV observations, and we have also validated the model in the work of La Spina et al. (2015)), one of the possible explanation of the reason we are not able to reproduce correctly the UV activity is related to one of the assumptions of the conduit model adopted: the 1D geometry. Indeed, with the presence of a shallow chamber a few hundred meters below the vent we can explain the high plagioclase content observed in the UV products. Magma rises from depth in disequilibrium towards the shallow chamber, it slows down its ascent (due to a change in decompression rate) and circulates within it for about 2 hours reaching the equilibrium crystal content, and then exits the chamber through a shallow conduit towards the surface. This idea is supported by the description of the plumbing system beneath UV reported by several authors (Behncke and Neri, 2003; Lanzafame et al., 2003; Metrich et al., 2004; Andronico et al., 2005; Neri et al., 2005; Allard et al., 2006; Corsaro et al., 2007; Bonforte et al. 2009; Sarao et al., 2010; Coulson et al., 2011; Corsaro et al., 2013; Gonzalez and Palano, 2014), making it reliable and reasonable.

A complete list of all the concerns and comments of the reviewers and the corresponding replies (in blue text) is reported below.

Reviewers' comments on manuscript NCOMMS-16-06494.

Reviewer #1

Summary of the key results.

The key result of identifying disequilibrium silicate melt with respect to volatile and crystal content driven by relatively rapid eruption rate is important. The results of the numerical simulations are tested against the properties of erupted materials and are consistent with existing hypothesised eruption scenarios and plumbing system structures.

Originality and interest.

This work is interesting and original, representing (to my knowledge) the first instance of coupling flow to phase (dis)equilibria: an important contribution.

Data & methodology: validity of approach, quality of data, quality of presentation.

The approach appears valid. It was not clear to me just how coupled the flow model was to the system disequilibrium and some clarification here would be useful. For instance, if the volatile phase was supersaturated was there less flow driving force?

The data are of high quality and the presentation good. I make comments on the attached annotated files.

Reply:

In order to make much clearer the coupling of the flow model with the disequilibrium processes, we have reported in the Method Section the entire description of the model, describing all the governing and constitutive equations adopted.

Regarding the question about the relation of the supersaturation of the volatile phase with the flow driving force, since all the variables within the model are coupled each other with highly non-linear relation, it is extremely difficult to predict with intuition alone the behaviour of the flow

for some initial condition. So, in order to answer this question, we have performed a simulation assuming supersaturation of the melt. First of all, the model assumes that, at the inlet of the conduit, the dissolved volatile content follows the equilibrium, therefore we cannot start the simulation assuming supersaturation condition. This is related to the fact that magma that enters the conduit comes from a magmatic chamber (just below the inlet), where we expect it had a long residence time and, therefore, reached the equilibrium conditions. However, posing a finite-rate exsolution (and an instantaneous crystallisation) we can simulate the supersaturation of the dissolved gas content along the conduit. Numerical results from this simulation show an increase (of a factor of 2) in the volume flow rate compared to the solution obtained assuming equilibrium exsolution. So against what seems intuitive, if the melt was supersaturated we would observe a lower resisting force, resulting in an increase in the volume flow rate. This result is related to the crystallisation, which is mostly controlled by the dissolved volatile content. Indeed, if the dissolved volatiles are not able to exsolve, the crystallisation is strongly limited, resulting in a decrease in the relative viscosity compared to the instantaneous exsolution case, and therefore in a decrease in the viscosity of the magmatic mixture. Finally, the decrease in the viscosity of the magma results in a greater volume flow rate, balancing the less driving force that would be produced by the less exsolved volatiles content related to the supersaturation condition.

Appropriate use of statistics and treatment of uncertainties.

A statistical approach is probably not appropriate here. Uncertainties are likely to be significant and are discussed. Nevertheless, the uncertainties appear small enough to allow discrimination between equilibrium systems and systems with significant dis-equilibrium.

Reply:

We agree with the reviewer that a statistical approach is not appropriate here.

Conclusions: robustness, validity, reliability.

The conclusions are robust in terms of demonstrating equilibrium from disequilibrium behaviour. However, I am concerned that the link to the 'effusive explosive transition' is stated but not discussed or demonstrated. This is important because it is the 'handle' on which the paper is sold, but the key findings remain important.

Reply:

The reviewer is right; we had not included a proper discussion on the relation between the crystallisation rate and the effusive-explosive transition in basaltic eruption. We have now inserted in the text a discussion on why the disequilibrium crystallisation and exsolution could be a key factor controlling the transition in eruptive style (see the detailed comments below). Our numerical results show that an increase in magma volume flow rate (for example due to perturbations in the initial conditions) results in an increase in gas volume fraction in the upper part of the conduit, that could lead, eventually, to magma fragmentation. Our model, however, does not consider (yet) the fragmentation of magma, therefore we cannot demonstrate properly this hypothesis.

Therefore, as suggested by the reviewer, we have focused the attention more on the key findings of the paper, rewriting the abstract and changing the title of the manuscript as follows:

“Key role for syn-eruptive plagioclase disequilibrium crystallisation in basaltic magma ascent dynamics”

Suggested improvements: experiments, data for possible revision.

References: appropriate credit to previous work?

Clarity and context: lucidity of abstract/summary, appropriateness of abstract, introduction and conclusions.

I have annotated my detailed comments on to the attached pdf file.

SJL

Other comments of Reviewer #1 annotated on the Manuscript

Manuscript (lines 10-12):

Rapid magma depressurization occurring during eruptions will limit the time available to crystallize minerals, thereby reducing magma viscosity and promoting even faster ascent.

Rev. #1:

Magma viscosity is not reduced.

If water is degassed then melt viscosity increases. If the volume proportion of crystals increases then the rheology responds with an increase in apparent viscosity and possible non-Newtonian behaviour.

I think what is meant here is that the magma viscosity is lower where the crystal content is less than the equilibrium value because of system dynamics.

Reply:

Yes, we agree with the reviewer. What we meant here is that due to the rapid magma depressurisation and the low viscosity, the ascent of a basaltic eruption is really fast. As a result, the time available for the crystallisation is very short, making the crystals not able to reach the equilibrium value before being erupted. Since the lower the crystal content the smaller the relative viscosity, the resulting viscosity of the basaltic magma during ascent is lower compared to that obtained assuming equilibrium crystallisation. Anyway, this sentence is no more present in the abstract, since, following the suggestions of all the reviewers, it has been completely rewritten.

Manuscript (line 21):

[...] of the conduit itself [...]

Rev. #1:

Not 'of the conduit', but 'of the conduit contents'. There is, of course, then an interaction between the mechanics of the conduit and the pressure within the magma fill.

Reply:

We have changed the text accordingly to the reviewer's suggestion.

Manuscript (lines 27-29):

However, it is now recognized that basaltic magmas rise fast enough^{1,11} that there is a limited time available for crystal growth and volatile exsolution, dramatically altering their viscosity and eruptive behaviour compared with an equilibrium ascent.

Rev. #1:

However, it is now recognized that basaltic magmas rise fast enough^{1,11} that the time available for crystal growth and volatile exsolution is minimal; this results in significantly different magma rheology and eruptive behaviour compared with an equilibrium ascent.

Reply:

We have changed the text accordingly to the reviewer's suggestion.

Manuscript (line 30):

[...] time constants [...]

Rev. #1:

'timescales', to be consistent with above.

Reply:

We have changed the text accordingly to the reviewer's suggestion.

Manuscript (lines 37-39):

Similarly, equilibrium CO₂ and H₂O contents are calculated using VolatileCalc¹³ for the P, T profile of magma ascent, and the exsolution time constant determines how swiftly equilibrium gas exsolution is achieved.

Rev. #1:

Question. Is there coupling between magma ascent rate and the gas exsolution that drives much of the ascent rate, particularly in the shallow conduit where the effusive - explosive transition takes place?

Reply:

As stated above, since all the equations of the model are coupled one to the other, it is not easy to determine exactly what is the main parameter driving the ascent rate. In particular, magma ascent rate and gas exsolution are clearly two-way coupled. Indeed, gas exsolution influences the crystal content, which affects the viscosity of the magma and ultimately the magma ascent rate. On the other hand, the increase in the magma ascent rate results in a decrease of the ascent time, which in turn produce a decrease of the exsolved gas content (dissolved volatiles has less time to reach the equilibrium profile).

In order to underline that the processes involved in ascent dynamics are strictly related one to the other we have added the following sentences to the text:

Lines 24-25 Revised Manuscript:

"Due to the highly non-linear interdependent processes involved in magma ascent dynamics, several simplifications are usually assumed when modelling their ascent."

Lines 39-42 Revised Manuscript:

"To study in detail the non-equilibrium crystallisation and exsolution, we use a 1D multi-phase multi-component steady-state model for magma ascent⁴, in which the main physical and chemical processes (such as crystallisation, exsolution, rheological variations, outgassing, non-ideal gas behaviour and temperature changes) are considered (see Methods Section)."

Manuscript (line 40):

[...] shallow [...]

Rev. #1:

Defined as (numbers)?

Reply:

In basaltic eruptions, plagioclase crystallises at relatively shallow depths (<10 km) (Francalanci et al 2004; Di Carlo et al 2006; Johnson et al 2008; Pichavant et al. 2009; Agostini et al. 2013; Lanzafame et al. 2013; Giacomoni et al. 2014; Mollo et al. 2015; Vetere et al. 2015). The main factors controlling plagioclase crystallisation are the pressure, the water content and the temperature of the magma. Plagioclase crystallisation is dependent on melt dissolved water content, therefore, it sensitive also to the exsolution characteristic time. Depending on these parameters, crystallisation of plagioclase can begin at different depths in the magma plumbing system. Considering a basaltic system characterized by water-saturated conditions and temperature of ~1100 °C, plagioclase is able to crystallise from ~50 MPa (~1.5 km below the vent) to the ambient pressure (Di Carlo et al 2006; Pichavant et al. 2009; Agostini et al 2013; Arzilli et al 2015). Textural and chemical information can be recorded by the crystallization of plagioclase in syn-eruptive conditions during the magma ascent at shallow depths (from ~2 km to the surface), therefore, plagioclase can act as a sensitive indicator of disequilibrium processes.

Accordingly to the suggestion of the reviewer we have modified the text as follows:

Lines 53-58 Revised Manuscript:

“In basaltic eruptions, plagioclase typically crystallises at relatively shallow depths (<10 km)^{20,26-28}, recording information of the magma ascent in the last kilometres below the vent. Considering a basaltic system characterised by water-saturated conditions and temperature of ~1100 °C, plagioclase is able to crystallise from 75-50 MPa to the ambient pressure^{20,27-29}, recording textural and chemical information of syn-eruptive conditions close to the vent of the conduit (from ~3 km to the surface). Therefore, plagioclase can act as a sensitive indicator of disequilibrium processes.”

Manuscript (line 41):

[...] where magma ascent is fastest [...]

Rev. #1:

This is likely to be in the region shallower than 100 m, does this correspond to the 'shallow' crystallization depth range?

Reply:

We agree with the reviewer that the region where the magma ascent is the fastest among the overall conduit is just a few hundred meters below the vent. Here we wanted to say the plagioclase is an indicator of the ascent dynamics in the last part of the conduit, i.e. from a few km below the vent to the surface. This sentence has now been rewritten as indicated in the comment above (see lines 53-58 Revised Manuscript)

Manuscript (lines 55-56):

We used a 1D multi-phase multi-component steady-state model for magma ascent⁴

Rev. #1:

Question 1. Does this model couple to the exsolution and crystallization dynamics to give instantaneous spatiotemporal rheology?

Question 2. How does a steady-state model adapt to interaction with non-equilibrium exsolution and crystallization?

Reply:

Question 1. Yes, of course. As reported in Fig. 1c, we can describe the spatial profile of the

viscosity of the magmatic mixture, which is determined by the pure melt viscosity (crystal-free and bubble-free) multiplied by a factor related to the presence of the crystal. The constitutive equations for magma rheology are now presented in the Methods Section of the revised manuscript. The evolution in time of all the parameters of the model (including the viscosity), instead, can be determined from the integration of the reciprocal of the mixture velocity all over the conduit, allowing us to see the behaviour in time of the variables of the model from depth to the vent of the conduit.

In order to make clearer that the model takes into account several processes, such as the rheological variations along the conduit, we have inserted the following sentence in the text:

Lines 39-42 Revised Manuscript:

“To study in detail the non-equilibrium crystallisation and exsolution, we use a 1D multi-phase multi-component steady-state model for magma ascent⁴, in which the main physical and chemical processes (such as crystallisation, exsolution, rheological variations, outgassing, non-ideal gas behaviour and temperature changes) are considered (see Methods Section).”

Question 2. The non-equilibrium crystallisation and exsolution in the model are taken into account through the Eqs. (5-8) described in Methods Section of the revised manuscript. These equations are part of the 1D steady-state model for magma ascent, as described in La Spina et al. (2015) and reported now in the Methods Section of the revised manuscript. At each depth, all the variables of the model are obtained in a way to solve simultaneously the system of equations (1-8) coupled with the constitutive equations (9-20). Therefore, when we set, for example, the characteristic time for exsolution $\tau^{(e)}$ to a very small value (with respect to the characteristic ascent time), we are forcing the term $(x_{d_i}^{md} - x_{d_i}^{md,eq})$ in Eqs. (5,6) to be also very small, which means that the dissolved volatiles content follows, at each depth, the equilibrium profile. If this happens, we say that the exsolution process is occurring instantaneously. On the contrary, if $\tau^{(e)}$ is not small enough, the model calculate the proper value of $x_{d_i}^{md}$ to balance Eqs. (5,6) (and all the others equations in the system). In this case, we say that we have a finite-rate exsolution. The same behaviour is valid also for the crystallisation process. Therefore, depending on the characteristic times and on the ascent time, the model computes a solution which follow the corresponding equilibrium profiles or not.

We have modified the manuscript as follows, clarifying how the finite-rate crystallisation and exsolution are taken into account:

Lines 90-91 Revised Manuscript:

“The 1D steady state model for magma ascent adopted here takes into account finite-rate crystallisation and volatile exsolution through the Eqs. (5-8) illustrated in the Methods Section.”

Manuscript (line 60):

[...] time constants for crystallisation, $\tau^{(c)}$, and exsolution processes, $\tau^{(d)}$.

Rev. #1:

Confusing superscript. Also 'd' = degassing or 'e' = exsolution? Be consistent.

Also degassing = exsolution + separation? i.e., degassing is not the same as exsolution?

Reply:

Here, the superscript “d” stands for “dissolved” and not for degassing. This notation was also adopted in the model described in La Spina et al 2015. However, following the suggestion of the reviewer, we have changed all the $\tau^{(d)}$ present in the text with $\tau^{(e)}$.

Manuscript (line 61):

~37%

Rev. #1:

What is the significance of 37%?

Reply:

This number comes from the definition of characteristic time (or time constants). The exact value is 1/e that is about 0.368. For simplicity, we have replaced “37%” with “1/e (~ 37%)”

Manuscript (line 63):

ϕ^{eq}

Rev. #1:

Why superscript? Subscript might be more conventional.

Reply:

Instead of the subscript, we used the superscript to be consistent with the other equilibrium values present in the Methods Section, i.e. β_j^{eq} and $x_{d_i}^{md,eq}$.

Manuscript (lines 64-65):

[...] same order as the magma ascent time, [...]

Rev. #1:

Is this the correct timescale? Volatile content is pressure dependent. Crystal content is volatile content dependent and pressure dependent. Maybe the rate of pressure change within a magma parcel is the key parameter, but maybe relative timescale follows from this?

Reply:

The timescale of magma ascent controls directly the decompression rate, therefore we use this timescale as comparison for the crystallisation and exsolution timescales. We have modified the text in order to underline that timescale of magma ascent controls the decompression rate:

Lines 96-98 Revised Manuscript:

“If these characteristic times are of the same order of magnitude or larger than the magma ascent time (which controls the decompression rate), disequilibrium processes will affect the ascent dynamics.”

Manuscript (lines 89-90):

[...] (potentially due to a different geometry).

Rev. #1:

Longer timescale storage in the central plumbing system?

Reply:

Our idea is that below UV is located a shallow magmatic chamber in which magma arrives in disequilibrium. Once magma reaches the shallow chamber, circulates within it in a convective flow (it is not just stacked there for a couple of hours) and, as suggested by the reviewer in one of the comments below, due to the increase in conduit-cross section we will have a

decrease in dP/dt . The crystals inside the magmatic mixture moving inside the chamber, have the time (that we have estimated to be ~ 2 hours) to reach the equilibrium before being erupted from UV.

To make clearer to the reader our interpretation of the plumbing system for both LV and UV (in agreement with the prevailing literature) we have modified the text as follows:

Lines 154-164 Revised Manuscript:

“Indeed, the equilibrium crystal content observed in UV can be explained with a longer ascent time than the LV, potentially due to a different complex geometry. Therefore, our results show that LV activity can be modelled with a non-equilibrium fast ascent from 9 km to the surface, whilst UV magmas suggest the presence of a shallow chamber or something similar in which magma can slow down its ascent (by decreasing the decompression rate dP/dt), circulates for a sufficient amount of time and reach equilibrium before being erupted. The high plagioclase content of UV products requires a low water content, and implies that this shallow reservoir should be located a few hundred meters below the vent. This is consistent with the prevailing literature, in which it is suggested that LV were fed by a straight vertical conduit (“eccentric” activity) whilst UV were fed a central conduit with a shallow horizontal dyke (“central-lateral” activity)^{31,32,36-46}”

Manuscript (lines 92-94):

Indeed, the decrease of the plagioclase content due to the disequilibrium processes causes a decrease of the viscosity, resulting in a difference of almost an order of magnitude at the vent of the conduit (3600 Pas vs. 500 Pas).

Rev. #1:

Is this order of magnitude difference due solely to the difference in plagioclase content, or a combination with higher melt water content (Fig. 1f, increased viscosity) and lower temperature (Fig. 1b, decreased viscosity)?

Reply:

Clearly, the higher water content dissolved and the lower temperature at the vent of the conduit observed in the non-equilibrium case play a role in the variation of the viscosity of the magma. The viscosity of the magmatic mixture is computed calculating the viscosity of the melt bubble-free and crystal-free, and then this value is corrected by the presence of the crystals. The viscosity of the bubble-free crystal-free melt is computed using the model of Giordano et al. (2008), whilst the model of Costa et al. (2009) has been adopted to correct the viscosity due to the presence of the crystals. The fitting parameters required by the model of Costa et al. (2009) are illustrated in La Spina et al. (2015). The viscosity of the pure melt is a function of the temperature and of the dissolved water content, whereas the relative viscosity depends only on the total crystal content. Assuming the instantaneous crystallisation and exsolution, the viscosity of the pure melt at the vent of the conduit is about twice the corresponding value obtained assuming the finite rates reported in the manuscript. The relative viscosity at the vent at equilibrium conditions, instead, is about 4 times greater than the value obtained from the results in disequilibrium. Therefore the lower temperature and the higher dissolved water content affected the final viscosity of a factor of 2, while the variations on the crystal content affected the viscosity of a factor of 4.

We have rewritten this sentence, underlying the fact that the variations in the viscosity is related not only to the decrease of the plagioclase content, but also to the water

supersaturation and the lower temperature:

Lines 130-132 Revised Manuscript:

“Indeed, the smaller plagioclase content, the melt supersaturation and the lower temperature due to the disequilibrium processes cause a decrease of the viscosity, resulting in a difference of almost an order of magnitude at the vent of the conduit (3600 Pa s vs. 500 Pa s).”

Manuscript (lines 95-97):

Our results show that LV activity can be modelled with a non-equilibrium fast ascent from 9 km to the surface, whilst UV magmas required shallow residence for a sufficient amount of time to reach equilibrium before being erupted.

Rev. #1:

This is consistent with the prevailing literature view?

Reply:

The prevailing interpretation of the plumbing system for the Etna 2001 flank eruption is consistent with ours. Most of the literature report that magma emitted from the LV ascended vertically from a deep reservoir that had likely been emplaced within the sedimentary substratum of Etna during the months preceding the eruption (Behncke and Neri, 2003). This part of the eruption was called “eccentric”. For the UV, instead, magma ascended from depth up to a few hundred meters below the Southeast Crater (SEC), and there it spread laterally, like in a horizontal dyke, towards the three summit vents located respectively at 2950 m, 2700 m and 2600 m (Behncke and Neri, 2003). For this reason, this part of the eruption was called “central-lateral”. Furthermore, assuming that the vent is located in the SEC (3050 m), we can estimate that the horizontal dyke is located between 100 and 500 m below the vent, consistent with our interpretation for the location of the shallow chamber below the UV.

The eccentric and lateral eruptive system of the 2001 flank eruption was supported by several authors, among which:

Behncke and Neri, 2003, B. Volcanol.
Lanzafame et al., 2003, J. Geol. Soc. London
Metrich et al., 2004, Earth Planet. Sc. Lett.
Andronico et al., 2005, B. Volcanol.
Neri et al., 2005, J. Volcanol. Geoth. Res.
Allard et al., 2006, Earth-Sc. Rev.
Corsaro et al., 2007, B. Volcanol.
Bonforte et al. 2009, Tectonophys.
Sarao et al., 2010, Geophys. J. Internat.
Coulson et al., 2011, Lithos
Corsaro et al., 2013, J. Volcanol. Geoth. Res.
Gonzalez and Palano, 2014, J. Volcanol. Geoth. Res.

We have edited the text more emphasising that our results are consistent with the prevailing literature (see above, lines 162-164 Revised Manuscript)

Manuscript (line 113):

[...] 20 min [...]

Rev. #1:

Unit of time in Fig 2? Or do you mean 1200 s?

Reply:

We have forgotten to put time units Fig. 2, and we thank the reviewer to let us notice this. The units are seconds, and the scale of the plots in Fig. 2 are all logarithmic. We have modified Fig. 2, adding the missing units.

Manuscript (lines 116-117):

[...] from 2 (with a long time constant for degassing) to 100 min.

Rev. #1:

What is the unit of time in Fig. 2?

Reply:

As above.

Manuscript (line 124):

[...] shear [...]

Rev. #1:

Equates to advection of crystallizing species?

Reply:

We apologise, but here we had to write “strain rate” and not “shear rate”. Indeed, with our 1D model we cannot take into account shear’s effects, but only the strain rate. However, we have removed this part of the sentence, rewriting it as follows:

Lines 194-198 Revised Manuscript:

“This timescale is faster than laboratory results as the natural system includes the combination of several processes, which are not simulated in the experiments. Our results, instead, include all the main processes which can control crystallisation rate, such as variations in undercooling, viscosity, strain rate and degassing, making the quantification of the characteristic time for plagioclase crystallisation more realistic.”

Manuscript (line 132):

[...] controlling transitions in eruption style.

Rev. #1:

The link from dynamic magma 'state' to eruption style has not been made.

Reply:

As we have discussed above, we agree with the reviewer that the link between non equilibrium crystallisation and exsolution and the transition in eruption style has not been fully demonstrated. Comparing the results at equilibrium and disequilibrium crystallisation and exsolution, we notice a larger gas volume fraction (in the upper part of the conduit) in the solution at disequilibrium with respect to the equilibrium case. In particular, at the vent of the conduit, we have obtained 81 vol.% against the 76 vol.% of exsolved gas respectively at disequilibrium and equilibrium crystallisation and exsolution. Therefore, if the model had considered a volume fraction criterion for the fragmentation (with a threshold of 80 vol.%), we would have obtained, in the case of disequilibrium crystallisation and exsolution, an explosive eruption. Furthermore, a variation in the volume flow rate (for example due to a perturbation in the initial conditions) would produce a change in the degree of

disequilibrium, varying also the gas volume fraction within the conduit. Therefore, the combination of disequilibrium processes and magma ascent rate could be a key factor controlling the transition in the eruptive style. We understand, however, that, this is not sufficient to demonstrate our hypothesis.

Therefore, we accordingly to what described above, we have modified the text as follows:

Lines 132-144 Revised Manuscript:

“The eruption rates derived from the numerical simulations are $\sim 8 \text{ m}^3 \text{ s}^{-1}$ for the solution at instantaneous crystallisation and exsolution, whilst $\sim 11 \text{ m}^3 \text{ s}^{-1}$ for that obtained assuming a finite-rate, both in agreement with real observations³⁵. Furthermore, the gas volume fraction obtained assuming finite-rate crystallisation and exsolution is larger (in the last 1.5 km below the vent) than that derived from the simulation at instantaneous condition. In particular, at the vent of the conduit, we obtained 81 vol.% against the 76 vol.% of exsolved gas respectively at disequilibrium and equilibrium crystallisation and exsolution. Thus, even though the model does not take into account fragmentation, if we had considered a volume fraction criterion for the fragmentation of magma with a threshold at 80 vol.%, the solution at disequilibrium crystallisation and exsolution would have produced an explosive eruption. Therefore, a variation in the magma ascent rate (for example due to perturbations of the initial condition) could change the degree of disequilibrium crystallisation and exsolution, producing, eventually, a change in the eruptive style.”

Manuscript (lines 134-136):

[...] resides within it for at least 1-2 hours reaching the equilibrium crystal content and then rises through the upper part of the conduit again in disequilibrium conditions.....

Rev. #1:

I'm not a fan of the stop-start idea. More satisfactory is the previously stated geometric idea where changes in conduit cross-section result in changes in dP/dt .

Reply:

In this sentence, the word “resides” is misleading. As stated above, our idea is that magma circulates within the shallow chamber of a certain amount of time before being erupted. As suggested by the reviewer, once a certain amount of magma enters the shallow chamber it slows down the ascent (due to the increase of the radius of the chamber compared to that of the conduit and resulting, thus, in a change of dP/dt) and circulates within it, due to a convective flow. Therefore, the magma, which has just entered the shallow chamber, takes some time before exiting the chamber itself, giving enough time to the crystals to reach the equilibrium content. Therefore, we have replaced the word “resides” with “circulates” (Line 159 Revised Manuscript)

Manuscript (lines 172-174):

Viscosity can vary by almost an order of magnitude between a full equilibrium effusive eruption (3600 Pas) and fast ascent explosive eruption (500 Pas) due to the impact of disequilibrium crystallisation.

Rev. #1:

I don't think that this correlation has been demonstrated. It seem to me more likely that if disequilibrium systems are more explosive then this is most likely due to the increased volatile content (Fig 1f), or at least a combination of volatile content and increased crystal content. Increased crystal content as a sole cause does not seem plausible.

Reply:

The change in the viscosity that we have observed is not only related to the impact of disequilibrium crystallisation, as previously written in the sentence taken into account, but also on the disequilibrium exsolution. Furthermore, we understand that the correlation between the variation of viscosity and changes in eruptive style is not well demonstrated here, therefore we have rewritten the sentence, just reporting what we have obtained from the numerical simulation:

Lines 245-247 Revised Manuscript:

"Furthermore, numerical results have shown that viscosity can vary by almost an order of magnitude between a full equilibrium eruption (3600 Pa s) and a non-equilibrium eruption (500 Pa s) due to the impact of disequilibrium crystallisation and exsolution."

Manuscript (line 175):

[...] controlling transitions in eruption style.

Rev. #1:

I am happy that the disequilibrium changes in volatile and crystal content have been plausibly demonstrated, but there is no discussion of how this then links to controlling effusive - explosive transitions; just stating this is not sufficient.

Reply:

As already stated, we agree with the reviewer that the link between non equilibrium crystallisation and exsolution and the transition in eruption style has not been formally demonstrated. Therefore we have rewritten the sentence in a more hypothetical way, including the relation with the variations in gas volume fraction we have discussed above:

Lines 242-244 Revised Manuscript:

"The timescale of magma ascent and disequilibrium processes could control also the transition in eruptive styles, since an increase in magma ascent rate produces an increase in the gas volume fraction in the upper part of the conduit, resulting, eventually, in the fragmentation of magma."

Manuscript (line 321):

Figure 2: Sensitivity analysis on the time constants.

Rev. #1:

You give the crystal content units, but not the timescale units: I presume seconds?

Reply:

Yes, the reviewer is right. We have corrected the figure inserting the units.

Reviewer #2

A. Summary of the key results

The paper under review considers disequilibrium crystallization of basaltic magma during ascent of the conduit. The main outcome of the modelling is that in order to explain Etna's product crystallinity a shallow reservoir located a few hundred meters below the summit should be

considered. This statement is not really well supported by the conduit flow model (see below).

Reply:

Our conclusion stating that in order to explain UV crystal content we need a shallow reservoir is not a direct result of the 1D conduit flow model. Using the 1D conduit flow model we have derived estimations for the characteristic times of crystallisation and exsolution, which give us solutions in agreement with the LV observations. Then, assuming that these characteristic times do not change with the plumbing system, we have concluded that these estimations should be valid also for the UV. However, the numerical results obtained assuming finite-rate crystallisation and exsolution are not in agreement with the UV observations (in particular due to the plagioclase content). On the contrary, if we assume instantaneous crystallisation and exsolution we are able to reproduce the UV observations, indicating that the UV products are able to reach equilibrium before being erupted. However, looking more in detail on the results, the numerical solutions in agreement with UV observations show an ascent time of less than 1 hour, too short to be consistent with the equilibrium crystallisation during a fast vertical ascent. Assuming that the model is describing properly the ascent dynamics of the eruptions (since we were able to reproduce correctly the LV observations, and we have also validated the model in the work of La Spina et al. (2015)), one of the possible explanation of the reason we are not able to reproduce correctly the UV activity is related to one of the assumption of the conduit model adopted: a continuous and uninterrupted vertical ascent from depth to the vent. Indeed, with the presence of a shallow chamber a few hundred meters below the vent we can explain the high plagioclase content observed in the UV products. Magma ascends from depth in disequilibrium towards the shallow chamber, it slows down and circulates within it for about 2 hours reaching the equilibrium crystal content, and then exits the chamber through a shallow conduit towards the surface. This idea is supported by the description of the plumbing system beneath UV reported by several authors, making it reliable and reasonable. Therefore, with our model we are not able to demonstrate directly the presence of a shallow chamber beneath the UV, but, thanks to it, we can demonstrate that the UV plumbing system is not a straight vertical conduit (like we have for LV) and that magma from UV is able to reach equilibrium before being erupted. Our findings suggest us the presence of a shallow chamber below the UV, hypothesis that is consistent with prevailing studies on the UV plumbing system.

B. Originality and interest: if not novel, please give references

Results are original and interesting in a sense that multiple crystal species are involved in the simulation. There was no previous non-equilibrium conduit model that accounts for several minerals crystallization. Authors completely ignore papers by Melnik, Sparks and Costa that established the link between ascent rate and crystallinity during lava dome building eruption. The transition between explosive and effusive eruption is not discussed in the paper at all. Key references to Slezin (1983 and 2003) are missing. There is no link between proposed crystal growth kinetics and experimental data on crystal growth in basaltic magmas.

Reply:

We apologize to miss these papers in the previous version of the paper, these studies are cited in the revised version of the manuscript:

Lines 24-30 Revised Manuscript:

“Due to the highly non-linear interdependent processes involved in magma ascent dynamics,

several simplifications are usually assumed when modelling their ascent. Indeed, initial conduit models assumed a single gas phase, isothermal conditions, no gas-magma separation and crystal-free magma^{5,6}. Later models reduced these simplifications, introducing gas-magma separation⁷⁻⁹, different volatile species^{10,11} and a total crystal phase (not distinguishing the mineral species)¹²⁻¹⁵. Afterwards, this single crystal phase was treated separately as microlites and phenocrysts¹⁶⁻¹⁸, but still as a single mineral."

Lines 39-45 Revised Manuscript:

"To study in detail the non-equilibrium crystallisation and exsolution, we use a 1D multi-phase multi-component steady-state model for magma ascent⁴, in which the main physical and chemical processes (such as crystallisation, exsolution, rheological variations, outgassing, non-ideal gas behaviour and temperature changes) are considered (see Methods Section). In this model, three different crystal components (plagioclase, clinopyroxene and olivine) and two different volatile species (water and carbon dioxide) are taken into account. This model is the first non-equilibrium conduit model that account for the crystallisation of different minerals."

Furthermore, links between our model and experimental data on crystal growth in basaltic magmas are introduced in the manuscript:

Lines 179-183 Revised Manuscript:

"This is a reasonable assumption since results from ref. 47 show that vesiculation of basaltic magmas happens in timescales of tens of seconds, whilst data from ref. 28 and 29 indicate that equilibrium crystallisation is reached in about two hours. Furthermore, the growth rates of vesicles (10^{-4} - 10^{-2} cm s⁻¹)^{47,48} are several order of magnitude higher than those of plagioclase (10^{-7} - 10^{-8} cm s⁻¹)^{28,29}."

C. Data & methodology: validity of approach, quality of data, quality of presentation

The model assumes single characteristic disequilibrium timescale for all crystal types and for all P-T condition. This assumption cannot be justified both experimentally and theoretically. Magmatic undercooling that drives crystallization is a complicated function of magma ascent P-T-t path and residual melt composition. Assumption of the constant delay time oversimplifies the system significantly. Thus, as a theoretical approach the model is valid but it cannot lead to conclusive data interpretation.

D. Appropriate use of statistics and treatment of uncertainties

Some parametric study is performed in the paper but simple variation of tau_c and tau_d does not lead to proper kinetic description of crystallization.

Reply:

We understand that the assumption of constant characteristic times is a simplification of the process. These characteristic time are reasonably function of the pressure, temperature, dissolved volatile content and bulk melt composition. However, at the moment, these functions do not exist, and in order to have a better description of the characteristic times, more laboratory experiments at disequilibrium conditions have to be conducted. In de' Michieli Vitturi et al. (2013), they presented numerical results from the investigation of several disequilibrium processes using a numerical 1D steady-state model applied to a rhyolitic eruption, validating the use of a constant

finite-rate exsolution against the data from Mangan and Sisson (2000) (see Fig. 1 reported below). Therefore, as first order approximation of the processes of disequilibrium crystallisation and exsolution, we have assumed that these characteristic times would be constant and the same respectively for each crystal phase and volatile component. The fact that we were able to reproduce the products observed from the LV suggests that our choice of characteristic times cannot be too far from the reality. Probably, during the ascent, the characteristic times vary all along the conduit going from a larger value at depth (where we have higher temperatures, small velocities, higher pressures), to a smaller value approaching to the vent (where we have lower temperatures, higher velocities and lower pressures). Our results suggest that the effects of the real non-constant characteristic times can be reproduced assuming sort of constant “effective” characteristic times, balancing their variation along the conduit.

Furthermore, in H₂O-saturated conditions and during decompression, characteristic time of crystallization could be considered constant, while, crystallization kinetics can change during the magma ascent. These evidences are experimentally proved by Agostini et al. (2013) and Arzilli et al. (2015), in fact, they show that the substantial amount of plagioclase crystallization can take place on short timescales (2 hours) at different final pressures, while, the nucleation and growth rates and the nucleation delay of plagioclase can change during decompression as a function of the undercooling. In other words, the undercooling is the driving force of the crystallization and controls the kinetics during the disequilibrium growth but the timescales to complete most of the crystallization could be similar at different pressures.

Therefore, we have modified the text as follows:

Lines 99-107 Revised Manuscript:

“Although a full description of the characteristic time for crystallisation and exsolution would require them to be functions of the pressure, temperature, dissolved water content and of the bulk composition of the melt, there are no experimental constraints on their values until now. In ref. 33, de’ Michieli Vitturi et al. presented numerical results from the investigation of several disequilibrium processes using a numerical 1D steady-state model applied to a rhyolitic eruption, validating the use of a constant finite-rate exsolution against the data from ref. 34. For this reason, in this work, as a first order approximation of the non-equilibrium crystallisation and exsolution in basaltic magmas, we assume that the characteristic times $\tau^{(c)}$ and $\tau^{(e)}$ are constant during the ascent and are the same respectively for each crystal phase and each volatile component.”

Lines 147-150 Revised Manuscript:

“From the numerical results presented here, we have shown that, although the characteristic times for crystallisation and exsolution are likely changing during the ascent, we are still able to reproduce the LV activity, and, therefore, at the first order, they can be approximated as some “effective” constant characteristic times.”

E. Conclusions: robustness, validity, reliability

The title of the paper is misleading. The paper is not about the transition between different eruption styles but about syn-eruptive crystallization.

Reply:

The reviewer is right, we have probably over emphasized the relation between the crystallisation rate and the effusive-explosive transition in basaltic eruption. We have discussed

above on the fact that a variation in the volume flow rate (for example due to a perturbation in the initial conditions) produces a change in the degree of disequilibrium, varying also the gas volume fraction within the conduit, leading, eventually, to magma fragmentation. Thus, our results suggest that crystallisation plays a fundamental role in controlling transitions in eruptive style, but we understand that, at this time, we do not have enough evidence to strongly support this hypothesis. Thus, following the suggestion of the reviewer, we have changed the title of the manuscript, focussing on the key findings of the paper:

“Key role for syn-eruptive plagioclase disequilibrium crystallisation in basaltic magma ascent dynamics”

F. Suggested improvements: experiments, data for possible revision

Crystal growth model should be significantly reworked based on available non-equilibrium crystal growth experiments.

Reply:

We appreciate the suggestion, but unfortunately few non-equilibrium crystal growth experiments at H₂O-saturated conditions, simulating decompression, are available in literature (Agostini et al. 2013; Arzilli et al. 2015) and they are not enough to rework the model. We just used the results of these experimental studies as a comparison with our numerical results. In the future, thanks to new NERC Large Grant DisEqm project (MB will be the PI), we will be able to perform a sufficient amount of crystallization experiments in basaltic melts in order to improve the crystal growth model.

G. References: appropriate credit to previous work?

See above

H. Clarity and context: lucidity of abstract/summary, appropriateness of abstract, introduction and conclusions

The abstract is not strongly linked with the main body of the paper.

Reply:

As we stated above, we have over emphasized in the title and in the abstract the relation between the disequilibrium crystallisation and the transition in the eruptive styles of basaltic eruption. We have, therefore, edited the abstract focussing more on the timescales of crystallisation that we have found from our results.

I suggest major revision of the manuscript.

Oleg Melnik

Reviewer #3

Review of La Spina et al. : Crystallisation rate controls hazardous effusive-explosive transitions in basaltic volcanism

Finite rate exsolution: $\tau_d = 10^4$, $\tau_c = +\infty$

A 8km length conduit (200 MPa at the inlet) has been used in these runs. Different values of the radius give different magma ascent rates and consequently different decompression rates.

This manuscript presents a kinetic framework to study the influence of disequilibrium crystallization and exsolution on the products (crystal abundances) and eruption style at Etna and some comparisons with Stromboli and Kilauea. The paper is generally easy to read and follow and pleasant to read. There are some very good ideas behind this study that I would like to see discussed and published. On the other end, there are some, for the most part cosmetic, suggestions I have to improve the experience for the readers and also one or two scientific concerns that I think can all be reasonably addressed by the authors. In the next paragraphs, I start with the positive points and then discuss some current aspects I am not entirely convinced about and finally suggest some more detailed changes for the authors.

The most interesting and significant contribution of the authors to me is the use of two independent variables to constrain a simple steady-state multiphase (1-D) conduit flow model, more specifically to constrain the ascent time of the magma and its departure from equilibrium. The kinetics of volatile exsolution and crystallization is assumed here to operate over slightly different time scales, which provides two constraints on the conduit dynamics for the ascent of magmas where information about glass water content (or vesicularity) ADD CAVEAT THAT FOR BASALTIC FLOWS>>> VESICULARITY AND CRYSTAL CONTENT CAN BE AFFECTED BY POST ERUPTION !! and crystal content (and assemblage) can be used to "invert" or fit for the best ascent scenario. This is a clever and elegant idea, the model proposed is simple and promising, although several underlying assumptions and implication of these assumptions are either glossed over or not really discussed. The discussion of the effect of the degree of disequilibrium on the phase assemblage is interesting and although not a new concept, the model offers some nice ways to test some interesting ideas about these eruptions. The authors have been working on related issues for a while and have shown in many instances that they produce high quality work.

I have some reserves first of all about the validity of the first order kinetics assumed for the exsolution and crystallization in the mass balance. Is there a way to validate or justify this assumption outside of the fact that it simplifies the analysis?

Reply:

The first order kinetics assumed for the exsolution has been validated against the data from decompression experiments performed by Mangan and Sisson (EPSL, 2000) for a crystal-free rhyolite melt. This validation has been presented by de' Michieli Vitturi et al. (EGU Meeting, 2013), showing that with an appropriate constant characteristic time for exsolution it is possible to fit the experimental data obtained at different decompression rates (see Fig. 1). For this reason, we have adopted the same first order kinetics for the exsolution (clearly with a different characteristic time) and we have extended it also to the crystallisation. We do not have yet enough non-equilibrium crystallisation and exsolution experiments to validate the first order kinetics assumed for basaltic magmas, but the fact that we are able to reproduce the observation from the LV activity suggest us that we are not too far from the reality.

In order to justify the reason why we have adopted constant characteristic times for crystallisation and exsolution, we have inserted the following paragraph to the text:

Lines 99-107 Revised Manuscript:

"Although a full description of the characteristic time for crystallisation and exsolution would require them to be functions of the pressure, temperature, dissolved water content and of the bulk composition of the melt, there are no experimental constraints on their values until now. In ref. 33, de' Michieli Vitturi et al. presented numerical results from the investigation of several disequilibrium processes using a numerical 1D steady-state model applied to a rhyolitic eruption, validating the use of a constant finite-rate exsolution against the data from ref. 34. For this reason, in this work, as a first order approximation of the non-equilibrium crystallisation and exsolution in basaltic magmas, we assume that the characteristic times $\tau^{(c)}$ and $\tau^{(e)}$ are constant during the ascent and are the same respectively for each crystal phase and each volatile component."

Moreover, I would have guessed that the "time constants" that the authors introduce in these first order reactions are probably not constants, but that the rate constants would depend on p,T and the magma composition, possibly even the flow conditions for exsolution (stretched bubbles have a greater Surface to Volume ratio so prone to grow faster). Because much of the paper revolves around the idea of these simple kinetic laws, it is worth spending more time discussing and justifying these choices.

Reply:

We agree with the reviewer that the time constants (or characteristic times) are probability not constant and they vary during the ascent. However, the fact that we were able to reproduce the products from the LV activity suggests that the estimations for the time constants we have found, at least at the first order, are representative of the real characteristic times during the ascent. Therefore, even though the real characteristic times are changing during the ascent, at the first order, they can be approximated with some "effective" constant characteristic times.

Furthermore, the timescales of crystallisation derived from the constraints on the characteristic time $\tau^{(c)}$ are in agreement with the timescales derived from laboratory experiments of Arzilli et al. (2015), suggesting that the "effective" characteristic times we have derived from the numerical simulations cannot be so different from the real characteristic times.

In order to have a better description of these characteristic times, we needed more laboratory experiments with basaltic samples in disequilibrium conditions. Unfortunately, these experiments are not easy not perform, explaining why there are just a few of them reported in the

literature. However, all the authors of this paper will be part a new NERC Large Grant DisEqm project (MB will be the PI), which the main aim is to quantify disequilibrium processes in basaltic volcanism. Within this project we will perform 4D X-Ray tomography experiments at high pressure and temperature to describe and model disequilibrium crystallisation during the ascent. These experiments will allow us to give better constraints on the characteristic times for the crystallisation and the exsolution process, but, at the moment, this is the best we can do.

As suggested by the reviewer, we have added some discussion about the choice of the constant characteristic times

Lines 147-150 Revised Manuscript:

“From the numerical results presented here, we have shown that, although the characteristic times for crystallisation and exsolution are likely changing during the ascent, we are still able to reproduce the LV activity, and, therefore, at the first order, they can be approximated as some “effective” constant characteristic times.”

The second point that needs to be discussed further is the constraint used to match or fit the most likely ascent scenario for each type of eruptions. Basaltic eruption products are known to be potentially affected by post-eruption processes, such as deformation upon landing and bubble collapse and partial volatile outgassing for explosive eruptions and crystallization and outgassing for lava flows. Are the constraints used for the analysis corrected for these effects? If so how?

Reply:

Corsaro et al. (2007) show that the crystallinity of tephra and lava flow erupted at the upper vent is similar, meaning that the lava flow products are not affected by post-eruption crystallization. Therefore, it is reasonable to assume a similar behaviour also for the LV, implying that the 2001 flank eruption at Mt. Etna was not affected by post-eruption crystallisation (or it was negligible).

The discussion on the issue on post-eruption crystallisation has now been included in the text:

Lines 81-85 Revised Manuscript:

“Furthermore, in ref. 31 it is shown that the crystallinity of tephra and lava flow erupted from UV is similar, meaning that the lava flow products are not affected by post-eruption crystallisation. Thus, it is reasonable to assume that even in the LV post-eruption crystallisation did not occur (or it was negligible), implying that the crystals observed in the erupted products were grown during the ascent.”

The third point concerns the UV eruption products and the short storage required at shallow depth to maintain close to equilibrium conditions at low pressure before the last ascent. The physical model that the author use does not allow for a "reservoir" perched in the middle or higher level of the conduit. If I understand well the model can play with different (but constant over depth) time constant and ascent rates, but most likely not with a region where the magma stalls. It is therefore misleading to show that the choice of time constants allows the authors to discuss the cases where the crystal content is high and a shallow storage is believed to exist.

Reply:

The reviewer is right, the model we are adopting assume a constant radius all along the

conduit, therefore we cannot consider a reservoir perched in the middle or higher level of the conduit. Thus we cannot demonstrate directly the existence of a shallow chamber in which magma circulates for a certain amount of time, enough to reach the equilibrium crystal content before being erupted. However, from our numerical simulations we can provide constraints on the characteristic times for crystallisation and exsolution comparing the results with observation from LV activity. Since the bulk composition of UV and LV products are quite similar, and during the 2001 eruption they were fed simultaneously, it is reasonable to assume that the characteristic times derived from the LV are valid also for the UV. However, using these characteristic times to simulate UV activity it is not possible to obtain an agreement with the observation, in particular for the plagioclase content. This allows us to say that the plumbing system for the UV cannot be a simple vertical conduit as we have for the LV (otherwise we would be able to reproduce the UV products just like we did for the LV). Therefore, there has to be at least a chamber located somewhere below the vent and the deep magma chamber (at 9 km). In the conduit-like part of the plumbing system, magma ascends in disequilibrium, controlled by the characteristic times derived from the LV simulations. Since plagioclase crystallise at shallow depth, if this chamber is located too deep with respect to the vent, plagioclase is not able to reach the high content observed from the UV products, therefore there has to be at least a shallow chamber (located a few hundred meters below the vent) where the P-T-H₂O conditions are suitable to crystallise the correct amount of plagioclase.

In conclusion our results suggest that below the UV is located at least a shallow chamber in which magma slows down its ascent and circulates for a couple of hours before being erupted, and the fact that other authors in the literature suggested also the existence of a shallow chamber below the UV makes our assertion more reliable.

Lines 150-164 Revised Manuscript:

“Moreover, it is reasonable to assume that they do not depend on the plumbing system, and therefore the characteristic times used to describe LV should be valid also for the UV. For this reason, since with a finite-rate crystallisation and exsolution we are not able to reproduce the observations from the UV, a possible explanation is that the assumption of a vertical straight conduit is not valid for the UV. Indeed, the equilibrium crystal content observed in UV can be explained with a longer ascent time than the LV, potentially due to a different complex geometry. Therefore, our results show that LV activity can be modelled with a non-equilibrium fast ascent from 9 km to the surface, whilst UV magmas suggest the presence of a shallow chamber or something similar in which magma can slow down its ascent (by decreasing the decompression rate dP/dt), circulates for a sufficient amount of time and reach equilibrium before being erupted. The high plagioclase content of UV products requires a low water content, and implies that this shallow reservoir should be located a few hundred meters below the vent. This is consistent with the prevailing literature, in which it is suggested that LV were fed by a straight vertical conduit (“eccentric” activity) whilst UV were fed a central conduit with a shallow horizontal dyke (“central-lateral” activity)^{31,32,36-46}.”

The authors also added a constraint on the time constant by assuming that the time for degassing is shorter than the time for crystallization, a justification for what seems mostly plausible is required here.

Reply:

In our model we assume that the time for exsolution is shorter than the time for crystallization. This hypothesis is supported by the results of Baker et al. (2012) in which they show

that vesiculation of basaltic magmas happens in the order of 10-20 seconds, whilst data from Agostini et al., 2013 and Arzilli et al., 2015 show that equilibrium crystallisation is reached in about two hours. Furthermore, the growth rates of vesicles range between 10^{-4} and 10^{-2} cm/s (Mangan et al., 1993; Baker et al., 2012), whereas, those of plagioclase crystals range between 10^{-7} and 10^{-8} cm/s (Agostini et al., 2013; Arzilli et al., 2015). These evidences show that the kinetics of vesicle are several orders of magnitude higher than those of plagioclase crystals, suggesting that the characteristic time for exsolution is smaller than the crystallisation one.

Following the suggestion of the reviewer we have modified the text as follows:

Lines 178-184 Revised Manuscript:

“Furthermore, we have limited our region of admissibility assuming that exsolution is a process occurring faster than crystallisation, i.e. $\tau^{(e)} \leq \tau^{(c)}$. This is a reasonable assumption since results from ref. 47 show that vesiculation of basaltic magmas happens in timescales of tens of seconds, whilst data from ref. 28 and 29 indicate that equilibrium crystallisation is reached in about two hours. Furthermore, the growth rates of vesicles (10^{-4} - 10^{-2} cm s⁻¹)^{47,48} are several order of magnitude higher than those of plagioclase (10^{-7} - 10^{-8} cm s⁻¹)^{28,29}. Therefore, these data suggest that the timescale of exsolution is shorter than the crystallisation one.”

Detailed comments:

- Title, although the effusive-explosive transition is part of the story, the title over-emphasizes the relevance of the present study to that question. The discussions with the viscosity feedbacks and other relevant aspect of the transition are relatively minor and not really the point of the paper. I would change the title.

Reply:

We agree with the reviewer. As stated above, we do not have yet enough evidence to strongly support the role of disequilibrium crystallisation on the transition in the eruptive style of a basaltic eruption. Therefore, we have changed the title focussing more on the role of the plagioclase disequilibrium crystallisation on the ascent dynamics.

- The reference to "time constants" knowing that the fact they are constants is merely an assumption is a bit weird. Could you refer to them as first order kinetic rates ($1/\tau$)? I think it is more clear and more precise.

Reply:

We adopted the terms “time constant” to indicate $\tau^{(c)}$ and $\tau^{(e)}$ since this is the definition used to indicate the parameter characterizing the response to a step input of a first-order, linear time-invariant system. If we consider the differential equation

$$\frac{dy}{dt} = -\frac{1}{\tau}y$$

with $y_0 = y(t=0)$ as initial condition, the solution is

$$y(t) = y_0 e^{-t/\tau}.$$

The time τ in these equations is called “time constant” or “characteristic time”, and this is why we used “time constants” to indicate $\tau^{(c)}$ and $\tau^{(e)}$ in our model. However, we agree with the reviewer that the use of “time constant” can lead people to consider them constant in time, assumption that we have really done to simplify the model, but that it could not be true in general. To avoid this confusion, we have replaced every instance of “time constant” with “characteristic time” that is still correct from a mathematical point of view and do not recall the idea of something that has to be constant in time. Furthermore, we agree with the reviewer that the use of constant characteristic times is just an assumption, and therefore we have referred to them as a first order approximation (see Lines 99-107 of the revised manuscript).

- Be careful in the log-log figures, your arguments have units (seconds for the time constants), this is not really clean. Could you remove dimensions before or at least mention that the units on these graphs are log(sec) - which is a bit weird but at least it is a bit cleaner.

Reply:

The reviewer is right, we have edited the figures in order to make clearer and consistent the units in the log-log figures.

- The radius of the conduit at Etna was chosen as 1 meter, that is very narrow... any justification for that (all the way down the 9kms)? What radius did you pick for Stromboli and Kilauea?

Reply:

We have chosen a radius of 1 meter for Etna performing several numerical simulations with different radius and comparing the resulting volume flow rate and the total crystal content with the real observations. The procedure is similar to that used in La Spina et al. 2015, where we have performed a sensitivity analysis on different initial conditions (in that case water content, temperature and conduit radius) deriving ranges of values for which the corresponding numerical solutions are in agreement with the observations. In this way, we have found that, with a radius of 1 m, we obtain a solution, at the equilibrium, in agreement with the UV, whilst assuming a finite-rate crystallisation and exsolution the corresponding solution is in agreement with the LV. In La Spina et al. 2015 we have found (for Stromboli) that an increase of 1 m in the conduit radius results in an increase of the mass discharge rate of more than 1 order of magnitude, and a similar result has been found for Etna, constraining, therefore, the range of admissible radii to no more than a few meters.

The results plotted in figure 3, where we have reported the plagioclase content as function of the volume flow rate, have been obtained performing many simulations with different radius. The range of radii used for Etna and Stromboli is 0.5-3.0 m, whilst for Kilauea we adopted 0.3-2.5m, and we can see that, increasing the radius of about 2.0/2.5m we have an increase in the volume flow rate of about 3 orders of magnitude.

We have now inserted in the description of the initial conditions (reported in the Supplementary Materials) the justifications described above.

- Line 124, you discuss how your model includes appropriate strain rate... well it is highly parameterized in a 1-D conduit model, you have some rough constraint on normal strain rates to some extent, but the shear rate not so much. I would be careful with that statement. Also the

realism of your plag crystallization model relies on the realism of 1st order kinetics, which is not really discussed (see comment above).

Reply:

As we stated above, here we had to write “strain rate” and not “shear rate”. Indeed, with our 1D model we cannot take into account shear’s effects, but only the strain rate. Anyway, we have removed the reference to the “shear rate” from the sentence.

Regarding the realism of the plagioclase crystallisation model, as we stated above, even though the real characteristic times are changing during the ascent, using a first order kinetics, we can approximate them as “effective” constant characteristic times. So, even though plagioclase is crystallising at each depth with different characteristic times (that is probably decreasing with the decrease of the pressure and the increase of the undercooling), we can approximate it with a constant mean value, and this is what we are doing in our model.

Since the main effect of the crystal content is related to the viscosity of the magma, our first order kinetics compared to the real crystallisation kinetics alters the viscosity along the conduit. However, due to the low plagioclase content (< 6 vol.% within the conduit), the variation of the viscosity related to the use of a constant characteristic time is small, and therefore it does not change particularly the ascent dynamics.

- Lines 133-136: Any geophysical or other evidence for the stalling of magmas at shallow depth in its path to the surface?

Reply:

Petrological evidences, mineral assemblage, plagioclase abundance and a combined thermodynamic and kinetic modelling of the compositional record of olivine crystals allow the constraining of a shallow reservoir between 75 and 5 MPa (Corsaro et al., 2007; Kahl et al., 2015). We have reported these evidences in the text:

Lines 162-166 Revised Manuscript:

“This is consistent with the prevailing literature, in which it is suggested that LV were fed by a straight vertical conduit (“eccentric” activity) whilst UV were fed a central conduit with a shallow horizontal dyke (“central-lateral” activity)^{31,32,36-46}. Furthermore, from petrological evidences, the shallow horizontal dyke should be located between 75 and 5 MPa^{31,46}, i.e. between ~2.5 km and the vent of the conduit.”

- The model and how it is setup is discussed very briefly and lacks details. I found the supplements on that account to fall a bit short. The discussion of the e-folding time for tau's is trivial but also requires on assumptions that are not relevant to the study (static magma...). That discussion to me should be replaced with a description of the model, initial and boundary conditions and sensitivity with initial conditions which are mostly missing. It would make the paper much more self-contained and pleasant to read.

Reply:

We agree with the reviewer, and thus we have reported in the Methods section a more detailed description of the model, with all the (governing and constitutive) equations.

- Why are the time constants fairly generic, i.e. applicable to other systems? The authors argue that it is controlled mostly by chemistry (thermodynamics), but the discussion there is a bit too succinct.

Reply:

The characteristic times are likely functions of pressure, temperature, water content and bulk composition of the melt, therefore, they are different for each volcanic system. So, in principle, we cannot use the same characteristic times for other volcanic system. However, if magmas have similar compositions, it is reasonable to assume the corresponding characteristic time quite similar to each other, at least at the first order. Therefore, we have used the “effective” characteristic times we have found for Etna, even for Stromboli and Kilauea, which have similar compositions but very different plagioclase content.

We have modified the text, including the justification reported above:

Lines 204-211 Revised Manuscript:

“As we stated above, characteristic times for crystallisation and exsolution are likely functions of the pressure, temperature, dissolved water content and of the bulk composition of the melt, and, therefore, they could vary with the different volcanic systems. However, for volcanic systems with a similar bulk composition it is reasonable to assume that the corresponding characteristic times are quite similar to each other, at least at the first order. Therefore, we have used the values for $\tau^{(c)}$ and $\tau^{(e)}$ deduced from Etna for other similar basaltic systems by performing numerical tests using magmatic compositions of Stromboli and Kilauea, whose products contain respectively high and low plagioclase contents.”

- Line 185-191: if you need some space, most of the sentences there could be removed or shortened. Hopefully, the readers should understand the difference between governing and constitutive equations.

- Line 195: how is the permeability parameterized and how does it affect your results? Much outgassing in your runs? There are no discussion on that anywhere else in the manuscript. I would write down the overall mass conservation for water (both dissolved and exsolved) or add the contribution of the mass balance for the exsolved part to illustrate that.

Reply:

The permeability was parameterised using the model described in Degruyter et al. (2012). However, with respect to that model, we have used the Darcy’s law, instead of the Forchheimer’s law. As said above, the complete description of the model is now present in the Methods Section.

- Line 218 and other places, please do not use "37%" in the text, which sounds arbitrary, rather used $1/e$.

Reply:

We have inserted the correct value $1/e$ in the text and we left in brackets the value $\sim 37\%$,

Final remarks: I enjoyed this manuscript and I think that if the authors can make respond and add some justifications about their model and address the main comments (plus cleanup some of the

details pointed just above), it will be worthy of publication in this journal. It was fun reading it, and reposed on several clever ideas.

Chris Huber

Reviewer #1 (Remarks to the Author):

Following significant revision from first review I have no further comments and recommend this manuscript for publication.

Reviewer #2 (Remarks to the Author):

The paper reads much better after a major revision of the text. I suggest that the paper can be published as it is except decreasing the number of insignificant digits in Table 2, and adding "for basaltic eruptions" in the first sentence of the paper because there are several non-equilibrium conduit flow models for silicic magma ascent. They are mentioned in the text.

Reviewer #3 (Remarks to the Author):

I read the responses to the reviewers comments and I think the authors have made a good job overall to address the issues. The manuscript is stronger now. Some of the additions (text) or changes may need further work to smooth the text a bit (grammar, awkward sentences). For example (not the only one), the new title, although more appropriate, sounds a bit odd... I would not start with "Key role of...". For instance, "The importance of..." is already a bit better. Other than that, I am favorable to publication. Best of luck !

Reviewers' and editor's comments on manuscript NCOMMS-16-06494A.

Reviewer #1

Following significant revision from first review I have no further comments and recommend this manuscript for publication.

Reviewer #2

The paper reads much better after a major revision of the text. I suggest that the paper can be published as it is except decreasing the number of insignificant digits in Table 2, and adding "for basaltic eruptions" in the first sentence of the paper because there are several nonequilibrium conduit flow models for silicic magma ascent. They are mentioned in the text.

Reply:

As suggested by the reviewer we have removed all the insignificant digits in Table 2, and we have added "for basaltic eruptions" in the first sentence of the paper (line 10 of the Revised Manuscript)

Reviewer #3

I read the responses to the reviewers comments and I think the authors have made a good job overall to address the issues. The manuscript is stronger now. Some of the additions (text) or changes may need further work to smooth the text a bit (grammar, awkward sentences). For example (not the only one), the new title, although more appropriate, sounds a bit odd... I would not start with "Key role of...". For instance, "The importance of..." is already a bit better. Other than that, I am favorable to publication. Best of luck!

C. Huber

Reply:

As suggested by the reviewer the text has now been revised. We have also modified the title as indicated by the editor in the attached manuscript. The title of the manuscript now reads: "Role of syn-eruptive plagioclase disequilibrium crystallisation in basaltic magma ascent dynamics"